# Transduction of Brain Neurons in Juvenile Chum Salmon (*Oncorhynchus keta*) with Recombinant Adeno-Associated Hippocampal Virus Injected into the Cerebellum during Long-Term Monitoring

**DOI:** 10.3390/ijms23094947

**Published:** 2022-04-29

**Authors:** Evgeniya V. Pushchina, Maria E. Bykova, Ekaterina V. Shamshurina, Anatoly A. Varaksin

**Affiliations:** A.V. Zhirmunsky National Scientific Center of Marine Biology, Far East Branch, Russian Academy of Sciences, Vladivostok 690041, Russia; stykanyova@mail.ru (M.E.B.); shamsh_gavrik@mail.ru (E.V.S.); anvaraksin@mail.ru (A.A.V.)

**Keywords:** recombinant adeno-associated viruses, chum salmon, cerebellum, intracerebroventricular administration, thalamo-cerebellar projections, green fluorescent protein, genetically encoded calcium indicators, Purkinje cells, eurydendroid cells, cerebrospinal fluid

## Abstract

*Corpus cerebelli* in juvenile chum salmon is a multiprojective region of the brain connected via afferent and efferent projections with the higher regions of the brainstem and synencephalon, as well as with multiprojection regions of the *medulla oblongata* and spinal cord. During the postembryonic development of the cerebellum in chum salmon, *Oncorhynchus keta*, the lateral part of the juvenile cerebellum gives rise to the caudomedial part of the definitive cerebellum, which is consistent with the data reported for zebrafish and mouse cerebellum. Thus, the topographic organization of the cerebellum and its efferents are similar between fish (chum salmon and zebrafish) and mammals, including mice and humans. The distributions of recombinant adeno-associated viral vectors (rAAVs) after an injection of the base vector into the cerebellum have shown highly specific patterns of transgene expression in bipolar neurons in the latero-caudal lobe of the juvenile chum *tectum opticum*. The distribution of rAAVs in the dorsal thalamus, epithalamus, *nucleus rotundus*, and pretectal complex indicates the targeted distribution of the transgene via the thalamo-cerebellar projections. The detection of GFP expression in the cells of the epiphysis and posterior tubercle of juvenile chum salmon is associated with the transgene’s distribution and with the cerebrospinal fluid flow, the brain ventricles and its outer surface. The direct delivery of the rAAV into the central nervous system by intracerebroventricular administration allows it to spread widely in the brain. Thus, the presence of special projection areas in the juvenile chum salmon cerebellum, as well as outside it, and the identification of the transgene’s expression in them confirm the potential ability of rAAVs to distribute in both intracerebellar and afferent and efferent extracerebellar projections of the cerebellum.

## 1. Introduction

Adeno-associated vector-mediated deoxyribonucleic acid (DNA) transfer is considered a promising avenue for targeted transgene delivery using intracellular transport [1,2]. One opportunity is retrograde transport, a fast axonal transport enhanced by dynein that allows transported agents to move towards the cell body from the axon ending. Physiologically, the retrograde transport delivers nerve growth factors from peripheral synapses to the cell body [3]. Studies show that some pathogenic viruses, such as herpes simplex virus [4] or poliovirus [5], use this pathway to enter neurons [6]. Based on this opportunity, some researchers use retrograde transport to introduce viral vectors into neurons through axons [7]. This issue was addressed for the mediator-specific nuclei of the ventral tegmental area (VTA) of mammalian midbrain in order to study the feasibility of targeted delivery of transgenes to remote areas of the brain [8]. Modern methods applied in such studies usually require several injections for the sufficient distribution of the vector to remote brain regions [8]. However, some types of recombinant adeno-associated viral vector (rAAV) transgenes can efficiently be transported via neuronal pathways associated with the injection site. Thus, genetic targeting in the central nervous system (CNS) can be achieved by scattering genes via long-axon antero- and retrograde projections from a limited area of the brain [9]. Studies of the dopaminergic VTA in mammals have shown a high efficiency of this methodology because the VTA is a region of the mesencephalon with numerous afferent and efferent projections to distant regions of the brain [8].

Previous studies on the tegmentum of juvenile chum salmon showed the ability of mammalian hippocampal rAAVs (GCaMP6m-GFP) to integrate into diencephalon and mesencephalon cells with a single injection, within one week into the area of the mesencephalic tegmentum [10]. In these studies, clusters of transgenic GFP-expressing nuclei (green fluorescent protein) and cells surrounded by GFP+ granules, with gradient distributions and forming discrete populations located in the area territory of the posterior tuberal nucleus, were identified in the chum salmon brain at one week after a single injection of GCaMP6m-GFP [10,11]. Thus, previous studies using confocal microscopy discovered the ability of rAAVs to integrate into the brain cells of juvenile chum salmon [12]. The results of short-term observations of the rAAV introduction into the chum salmon tegmentum region revealed traces of the vector genome’s transfer to regions of the brain that were sufficiently remote from the injection zone. However, to date, issues related to the potential delivery of genetic vectors remote from the injection site require further study of the mechanisms involved in transgenic vector transport to remote brain regions. The issue of phenotypes of transduced cells is also important, since many vectors are capable of infecting both glia and neurons [13]. Addressing these issues will help to identify the causes of different signal intensities in areas located at different distances from the injection zone, as well as the features of the vector’s intracellular metabolism. In particular, with regard to rAAV vectors, the issue of the vector’s capability of autonomous reproduction in the non-genomic DNA of the host cells as part of epysomes is of fundamental importance [14].

Data on the regional organization of the cerebellum in various fish species are mainly based on the anatomical mapping of axonal projections of cerebellar afferents [15]. Another approach is the study of regionalization of cerebellar efferents [15,16], electrophysiological approaches [17], and transneuronal tracing [15,18,19].

Modern non-invasive methods of in vivo functional regionalization can be considered one of the most interesting and promising approaches [15]. These methods were first applied to the zebrafish *Danio rerio*, whose regionalization of the cerebellar body was studied using genetically encoded transneuronal mapping of efferent chains of Purkinje cells (PC) [15]. It was shown that the functional chains of cerebellar cells, their afferents and efferents, correspond to the activity of PCs characterized by spatial specificity, and also determine the free behavior of zebrafish larvae during the performance of cerebellar-dependent behavioral acts [15]. Optogenetic studies of PC from selected areas of the cerebellum during animal behavior confirm the functional regionalization of PC efferents and determines their contribution to behavioral responses. Thus, the use of non-invasive methodology in such studies can clarify how different parts of the brain are associated with the performance of multiple functions, by highlighting the specialized efferent circuits for fulfilling certain behavioral tasks.

Another important issue is the suitability of rAAVs for gene therapy, compared to adenoviruses (ADV). Currently, the use of rAAVs for gene therapy to treat Parkinson’s disease (PD) is an extremely promising approach [1]. This is due to their considerable flexibility that allows them to affect several pathobiological processes, for which low-molecular-weight agents are not available. Genetic approaches to PD were evaluated in Phase I and II clinical trials [20,21]. These studies included both enzyme-based therapies aimed at relieving symptoms [22] and growth factor-based approaches with potential neuroprotective and neurorestorative effects [23]. The clinical approaches to gene delivery using viral vectors based on the adeno-associated virus (AAV) proved to be most effective in PD patients [1].

In fish brain studies, the use of rAAVs is often associated with the appearance of granules, with both intracellular and extracellular localization in projection targets [4]. To date, little is known about the process of further infection of cells after AAV attachment to the cell surface. The mechanism of penetration of most non-enveloped viruses is not well understood. Several different scenarios have been proposed based on morphological, ultrastructural, and biochemical studies [24]. Adeno-associated virus receptors (AAVR) have been identified as an important AAV receptor for several serotypes [25]. It is worth noting that AAVRs may play a major role in facilitating intracellular viral transfer [26]. Receptor-interacting regions of several serotypes have been identified, which together constitute the basis for the rational engineering of capsids with desired properties [27]. It has been shown that successful receptor recognition can lead to internalization by endocytosis [28]. Intact rAAV particles in endosomes undergo a series of pH-dependent structural changes required for transduction [29] and movement through the cytosol, via the cytoskeletal network [30]. After exiting the endosome, the rAAV enters the nucleus through the nuclear pore complex [31], where its capsid is cleaved to release the genome. It is important to note that the intracellular transfer involves multiple cellular events and can be interrupted at any stage, resulting in failed gene delivery. Therefore, the identification of key host cellular factors and the underlying mechanisms that regulate this process have great potential to increase the efficiency of rAAV transduction.

There is increasing evidence that host factors affecting the efficiency of gene delivery are triggered as soon as a rAAV is injected. For example, new data show that different AAV serotypes interact differently with serum proteins [24]. However, the efficacy of rAAVs is largely determined by molecular interactions between the capsid and the target cell surface receptors [32], with subsequent downstream events after particle internalization [33]. It is assumed that the predominant AAV serotypes recognize the different cellular receptors that contain glycoproteins in their structure and, therefore, exhibit different tissue and cell type tropism profiles [24]. The combinatorial recognition of coreceptors may also be involved in cell surface binding and internalization [33]. However, the ways in which AAV vectors enter cells, followed by translocation to the nucleus and localization within the cell so that their transgenes can be expressed, still remain unclear. Some biochemical and genetic data have been obtained that show the cell surface heparan sulfate proteoglycan (HSPG) to serve as the primary receptor for AAV attachment [25]. The fibroblast growth factor receptor [34] and avb5 integrin [35] are also involved as co-receptors or mediators of AAV entry into target cells.

Adv5-based vectors, after being introduced into the CNS, infect astrocytes to a greater extent than neurons [36]. In this regard, the tropism of Adv5-based vectors is determined by the direct binding of the Adv5 fiber protein to its primary cellular Coxsackie receptor (CAR) and adenovirus receptor [37]. The dependence of Adv5 on the transduction of CAR expression leads to a situation in which non-specific cells with a high content of CAR can be infected, while target tissues with a low level of CAR remain poorly infected [38]. Indeed, many clinically relevant tissues are often immune to Adv5 infection due to negligible levels of CAR, but tropism-modified Adv vectors that use alternative receptors can be successful [39].

The main objectives of this study are the long-term monitoring of rAAV (GCaMP6m-GFP) distribution to study, in particular, the potential of its transduction into the cerebellum cells of juvenile chum salmon, *Oncorhynchus keta*, and the ability to be transported as part of anterograde and retrograde projections, both inside the cerebellum and as part of remote transcerebellar projections.

## 2. Results

As a result of a single injection of the rAAV (GCaMP6m-GFP) into the dorsal part of the *corpus cerebelli* in juvenile chum salmon, rAAV transduction was identified in various parts of the diencephalon and mesencephalon. Intracerebellar dorsal, lateral, and basal zones containing GCaMP6m-GFP transduction were identified. In the mesencephalon, transcerebellar GCaMP6m-GFP transduction was detected in the latero-caudal part of the *tectum opticum*. In the diencephalon, transcerebellar GCaMP6m-GFP transduction was identified in the following parts of the synencephalon (optical nucleus of the thalamus): the *corpus geniculatum*, posterior tuberal region, dorsal thalamus, and epithalamus. The distribution pattern of GCaMP6m-GFP expression in the considered brain region is shown in Graphical abstract.

As a result of a single rAAV injection, both intracerebellar and extracerebellar areas of GCaMP6m-GFP transduction were identified. This probably indicates the presence of extracerebellar projections of the *corpus cerebelli* into the tectum. The transduction of GCaMP6m-GFP in the thalamic nuclei confirms the presence of tecto-cerebellar, thalamo-cerebellar, and tubero-mesencephalo-cerebellar projections in the salmon brain. The morphological and densitometric parameters of GCaMP6m-GFP-transducing cells and granules, HuCD-expressing neurons, and DAPI-stained nuclei are shown in Appendix A. Data on the quantitative ratio of HuCD-labeled and GCaMP6m-GFP expressing cells and nuclei, as well as their co-localization at 3 months after a single injection of the vector, are presented in Appendix A.

The results of the negative control, showing the lack of expression of the rAAV vector (GCaMP6m-GFP+ signal and HuCD+ signal), and also the results of double labeling without the presence of primary antibodies are shown in Figure 1.

### 2.1. Areas of Intracerebellar GCaMP6m-GFP Expression in the Juvenile Chum Salmon Cerebellum

#### 2.1.1. Dorsal Part of Corpus Cerebelli

DAPI staining in the dorsal cerebellum identified clusters of stained nuclei in the dorsal matrix zone (DMZ) (Figure 1A), and also densely stained clusters of cell nuclei in the apical zone (Az) of the *corpus cerebelli* (Figure 1A1,A2). A morphological analysis of the nuclei of these areas showed the presence of two types of small, rounded or oval nuclei in the DMZ (Figure 1A) and in the intermediate region between the DMZ and Az (Figure 1A, Appendix A). Larger, medium-sized nuclei dominated the middle part and, to a lesser extent, the dorsal part of the Az (Figure 1A1, Appendix A). The largest elongated nuclei with numerous small nucleoli were found in the Az (Figure 1A2, Appendix A).

The GCaMP6m-GFP expression in the dorsal cerebellum was detected in small and larger cells of the Az and in numerous granules (Figure 1B, Appendix A). The results of the negative control for GCaMP6m-GFP labeling in the dorsal zone are shown in Figure 2A. The intensity of the fluorescent signal in small cells was moderate, while in larger cells it was high (Figure 1B). Numerous diffusely arranged GFP+ granules exhibited intense fluorescence (Appendix A). No GFP expression was found in the DMZ. Immunohistochemical (IHC) labeling of the neuronal calcium-binding protein HuCD in the dorsal cerebellum revealed a few small, intensely labeled and/or larger moderately labeled neurons (Appendix A, Figure 1C). The results of the negative control for HuCD immunolabeling in the dorsal zone are shown in Figure 2B. In medium-sized and large cells (10.3 and 14.8 µm), the HuCD+ signal was visualized in the cytoplasm, with the nucleus being immunonegative; small cells were labeled completely (Appendix A, Figure 1C). When three optical channels were superimposed, GFP/HuCD co-localization in the dorsal part of the cerebellar body was detected in small cells, and the signal intensity was usually high (Figure 1D, Appendix A).

In these cells, DAPI-stained nuclei containing 1–2 nucleoli were observed (Figure 1D). Numerous GFP+ moderately labeled granules were arranged autonomously (Figure 1D). The results of the negative control for HuCD and GCaMP6m-GFP immunolabeling in the dorsal zone are shown in Figure 2C. The results of the one-way analysis of variance (ANOVA) for the distribution of GFP+, HuCD+, and GFP+/HuCD+-labeled cells are shown in Figure 1E. The results of the ANOVA indicated significant intergroup differences in the distribution of GFP+ and GFP+/HuCD+ cells (*p* ≤ 0.05). The percentages of GFP+ and GFP+/HuCD+ cells are shown in Appendix A.

#### 2.1.2. Lateral Part of Corpus Cerebelli

***HuCD expression in ganglion cells***. After staining with DAPI, in the molecular, granular and ganglionic layers of the juvenile chum salmon cerebellum, a different density of distribution of nuclei was revealed (Figure 3A). The ganglion layer of cells in the juvenile chum salmon cerebellum was formed by the following two types of projection neurons: pear-shaped Purkinje cells (PC) and eurydendroid cells (EDC), located between the molecular and granular layers of the *corpus cerebelli* (Figure 3B). These cells were examined in the lateral zone of the chum salmon cerebellum (Figure 3A, pictogram), where their topography and morphological features were mostly typical. The intensity of immunofluorescence (IF) HuCD labeling in PC and EDC of juvenile chum salmon varied (Figure 3B1). The distribution density of HuCD+ PC and EDC differed; in the lateral region, the cells were in clusters of 2–3, without forming a single layer (Figure 3B2). On the side of the granular layer, numerous DAPI-stained nuclei (Figure 3B2) were adjacent to HuCD+ PC and EDC. In some areas, the typical distribution pattern of HuCD+ PC and EDC changed, forming single clusters of HuCD+ PC and EDC containing DAPI-stained nuclei, and also proximal parts of HuCD+ ascending PC dendrites (Figure 3B3). However, the most typical distribution pattern was that composed of single HuCD+ bipolar EDCs and pear-shaped PCs, whose proximal HuCD+ parts of the dendrites were also identified in the ganglionic layer (Figure 3C).

In the lateral zone (pictogram, Figure 4A), DAPI staining revealed numerous clusters of small nuclei forming zones of reactive neurogenesis (Figure 4A, Appendix A). The nuclei were grouped into conglomerates of various sizes, including rather large and smaller clusters (Figure 4A). Intensive GFP expression in the lateral region was detected in three cell types (Figure 4B, Appendix A). Smaller and larger GFP+ expressing cells tended to be single and had intense GFP+ fluorescence (Figure 4B1). An analysis of HuCD immunofluorescence (IF) in the lateral zone of the *corpus cerebelli* showed the presence of two types of intensely labeled cells (Figure 4C, Appendix A). These cells had intense or moderate IF intensity and cytoplasmic localization of HuCD (Figure 4C1). With a moderate intensity of immunolocalization, a narrow-labeled rim of cytoplasm and the immunonegative nucleus were visualized (Figure 4C1). The three-channel scanning revealed co-localization patterns of DAPI, GFP, and HuCD (Figure 4E). Co-localization of HuCD/GFP with an intense signal was detected in two cell types of the lateral region (Figure 4D1, Appendix A). The results of the one-way ANOVA did not show significant intergroup differences in the distribution of GFP+, HuCD, and GFP+/HuCD+ signals (Figure 4E). The percentage distribution of GFP+ and GFP+/HuCD+ cells is presented in Appendix A.

#### 2.1.3. Basal Part of Corpus Cerebelli

In the basal part of the cerebellum, DAPI staining revealed numerous heterogeneous nuclei of four types, devoid of nucleoli (Appendix A, Figure 5A). The DAPI-stained nuclei were diffusely arranged, occasionally forming small dense conglomerates (Figure 5A1). GFP expression was detected in three cell types (Appendix A). Intense GFP labeling was characteristic of small, rounded cells (Figure 5B1). Larger cells had a moderate GFP expression (Appendix A). Moderate HuCD IF was detected in two cell types (Appendix A). Larger cells were characterized by intense IF (5C, inset); smaller cells were characterized by a moderate intensity of HuCD IF (Figure 5C1, Appendix A). A scanning through three channels in the basal zone revealed two types of cells containing a moderate or high intensity of co-localization of the GFP+/HuCD+ signal (Figure 5D). Along with cells containing co-localization, numerous single GFP-labeled granules were also identified. The results of the one-way ANOVA did not reveal intergroup differences in the localization of GFP+ and GFP+/HuCD+ cell groups (Figure 5E). The percentages of GFP+ and GFP+/HuCD+-labeled cells are presented in Appendix A.

### 2.2. Extracerebellar GCaMP6m-GFP Expression in the Tectum Opticum

DAPI staining of nuclei in the juvenile chum salmon *tectum opticum* revealed a weakly differentiated structural organization of this integrative center of the brain in juvenile fish (Figure 6A). The *tectum opticum* contained a large number of undifferentiated nuclei of small and larger sizes located in the tectal parenchyma, with larger nuclei occasionally found in the deep layers of the tectum (Figure 6A, Appendix A). The weakly differentiated structure of the tectum was supplemented by a large number of diffusely located nuclei belonging to cells in a state of migration, reactive neurogenic clusters, and single small undifferentiated nuclei, mainly with single nucleoli (Figure 6A1).

However, GFP expression was detected in the cytoplasm of elongated cells with bipolar morphology, located in the *stratum grizerum et album periventriculare* (SGAP). Weak to moderate GFP expression was also detected in the granules (Appendix A). Two types of elongated cells were characterized by intense or moderate IF intensity and a high level of cytoplasmic expression of GFP; in smaller cells, a moderate level was revealed (Figure 6B,B1, Appendix A).

As a result of the analysis of HuCD IF in the tectum of juvenile chum salmon, several cell types were identified, including vertical bipolar, pear-shaped, polygonal, and rounded cells (Figure 6C, Appendix A). Thus, the labeling of the neuron-specific HuCD protein made it possible to identify different morphological cell types in the *stratum grizerum centrale* (SGC) and SGAP, which indicates the neuronal differentiation processes in juvenile chum salmon of this age group. The results of a topographic analysis of the neuron-specific IF of cells in SGC and SGAP showed the presence of a heterogeneous cellular composition of HuCD+ cells (Figure 6C1). Four types of intensely labeled HuCD+ cells were identified (Appendix A). A scanning through three channels revealed co-localization patterns of GFP and HuCD in horizontal bipolar cells, located in the area of SGC and SGAP (Figure 6D). No co-locolization of these markers in other types of HuCD+ cells was detected (Figure 6D1). The one-way ANOVA showed no intergroup differences in signal distribution in GFP+, HuCD+, and GFP+/HuCD+ cells (Figure 6E). The percentage of GFP+ and GFP+/HuCD+ expressing cells is shown in Appendix A.

#### 2.2.1. Extracerebellar GCaMP6m-GFP Transduction in the Dorsal Thalamus

After a single injection of the rAAV into the cerebellum, GCaMP6m-GFP transduction was detected in the dorsal part of the diencephalon, in the following areas: the periventricular nuclei of the epithalamus, the nuclei of the dorsal thalamus (DTN), the nucleus rotundus, and the epiphysis. The presence of transgene expression in these areas of the juvenile chum salmon brain confirms the existence of antero/retrograde thalamo-cerebellar projections in salmonids.

The DAPI staining of the DTN cells (Figure 7A, pictogram) revealed morphologically heterogeneous, rounded and elongated nuclei with a large number of nucleoli (Figure 7A, Appendix A). In the DTN, GFP expression was detected in granules, as well as in small cells/nuclei (Appendix A, Figure 7B). The pattern of GFP expression in cells and nuclei varied between the periventricular zone (PVZ) and subventricular zone (SVZ) (Figure 7B1). Intense IF was detected in granules and PVZ cells (Figure 7B1). Less intense expression was observed in SVZ cells (Figure 7B1, Appendix A). Intense HuCD IF was detected in four cell types (Figure 7C, Appendix A). Localization of HuCD was observed, as a rule, in the cytoplasm of cells (Figure 7C). Three-channel scans revealed patterns of extensive distribution of GFP+ small cells and granules (Figure 7D). However, moderate/strong co-localization of GFP+ and HuCD+ signals was detected in a single cell type (Figure 7D1, Appendix A). The one-way ANOVA revealed significant intergroup differences in the distribution of GFP+ and HuCD+ cells (*p* ≤ 0.05) and GFP+ and GFP+/HuCD+ cells (*p* ≤ 0.05) and significant intergroup differences in the localization of HuCD+ and HuCD/GFP+ cells (*p* ≤ 0.01) (Figure 7E). The percentage of GFP+ and GFP+/HuCD+ cells is shown in Appendix A.

#### 2.2.2. Extracerebellar GCaMP6m-GFP Transduction in the Epithalamus

The DAPI staining of nuclei in the epithalamus clearly identified the PVZ and SVZ (Figure 8A). Among the stained nuclei in the PVZ, rounded or oval nuclei with a large number of nucleoli and small elongated nuclei without nucleoli were distinguished (Figure 8A1, Appendix A). In the SVZ, as a rule, oval nuclei containing one or two nucleoli were identified (Figure 8A1). Moderate or intensive GFP expression was detected in a heterogeneous population of cells and granules (Appendix A, Figure 8B). Small, intensely labeled GFP+ cells were found in the subventricular parenchymal part of the epithalamus nucleus. Larger, intensely labeled cells were found in the lateral population (Figure 8B2). As a result of the HuCD IF analysis, three types of HuCD expressing cells were identified (Figure 8C, Appendix A). In most cells, the cytoplasmic localization of the HuCD protein was recorded (Figure 8C1). Three-channel scans revealed discrete HuCD+ and GFP+ cell populations within the epithalamus (Figure 8D). Co-localization of HuCD/GFP signals was detected in two cell types (Figure 8D1, Appendix A). However, most epithalamic cells showed separate labeling of HuCD+ cells and GFP+ expressing granules. The one-way ANOVA for the distribution of immunolabeled cells revealed significant intergroup differences between the ratios GFP+/HuCD+ cells, as well as HuCD+ and GFP+/HuCD+ cells (*p* ≤ 0.01) (Figure 8E). The percentage of GFP+ and GFP+/HuCD+ cells is presented in Appendix A.

#### 2.2.3. Extracerebellar GCaMP6m-GFP Transduction in the Epiphysis

Numerous nuclei of neurosecretory cells were revealed in the structure of the epiphysis by staining with DAPI (Figure 9A, pictogram). In the region of the epiphysis, four types of DAPI-stained nuclei were identified, in which nucleoli were not visualized (Figure 9A1, Appendix A). In the epiphysis, GFP expression was detected in two cell types and granules (Figure 9B). Intensely labeled granules, as well as intensely or moderately labeled cells, formed an extended fragment in the superficial layers of the epiphysis (Figure 9B1). HuCD expression was detected in three types of neurons in the epiphysis (Figure 9C). Intense or moderate IF HuCD, as a rule, was observed in the cytoplasm (Figure 9C1). The scanning in three channels revealed heterogeneous patterns of localization of HuCD+ and GFP+ signals in the epiphysis (Figure 9D). Co-localization of GFP/HuCD signals was detected in two cell types located in the superficial part of the epiphysis (Figure 9D1, Appendix A). The one-way ANOVA for the distribution of immunolabeled cells revealed significant intergroup differences in the numbers of GFP+ and GFP+/HuCD+ cells (*p* ≤ 0.05) and significant intergroup differences in the distribution of HuCD+ and GFP+/HuCD cells (*p* ≤ 0.01) (Figure 9E). The percentage of GFP+ and GFP+/HuCD+ cells is presented in Appendix A.

### 2.3. Extracerebellar Transduction of GCaMP6m-GFP in the Posterior Tuberal Region

The posterior tubercle is the caudal boundary of the prosencephalon cell migration and is an eminence of the diencephalon, containing numerous poorly differentiated cell masses (Figure 10A, pictogram). The DAPI staining in the area of the caudal part of the posterior tubercle (PTp) revealed numerous clusters of small, undifferentiated nuclei of two types (Figure 10A, Appendix A). An analysis of nuclei from the PVZ and SVZ of the posterior tubercle showed the presence of small, poorly differentiated nuclei in the PVZ and SVZ and larger, elongated or oval nuclei located in the deeper parenchymal layers of the brain (Figure 10A1). The GFP labeling in PTp revealed small clusters of granules, as well as moderately labeled cells. The IF detection of HuCD showed the presence of three cell types (Figure 10C, Appendix A). Most IF cells had cytoplasmic localization of HuCD, but some small cells were completely labeled (Figure 10C1). The scanning in three channels identified discrete clusters of HuCD+ neurons in the SVZ, and local clusters of stained DAPI in reactive neurogenic niches (RNNs) and isolated GFP+ granules (Figure 10D, Appendix A). An examination under high magnification showed that some HuCD+ neurons also contained GFP+ granules (Figure 10D1, inset), but most GFP+ granules did not co-localize with HuCD.

The one-way ANOVA for distribution revealed the presence of intergroup differences in the proportions of immunopositive GFP+ and GFP+/HuCD+ cells (*p* ≤ 0.05), as well as significant intergroup differences in the distribution of HuCD+ and GFP+/HuCD cells (*p* ≤ 0.01) (Figure 10F). The percentage of GFP+ and GFP+/HuCD+ cells is presented in Appendix A.

### 2.4. Extracerebellar Transduction of GCaMP6m-GFP into the Nucleus Rotundus

The anterior thalamic nucleus or glomelular nucleus (*nucleus rotundus*) is the largest nucleus of the thalamus, including the cellular and neuropil components. Numerous small nuclei were found in the *nucleus rotundus* of juvenile chum salmon (Figure 11A, pictogram). The DAPI staining in the *nucleus rotundus* revealed a complex diffuse pattern of nuclear organization, including migrating populations (Figure 11A). An examination at higher magnification revealed an accumulation of nuclei containing numerous nucleoli in the *nucleus rotundus* (Figure 11A1). GFP expression was detected in several cell types and granules (Figure 11B). Patterns of granular expression of varying intensity were found in the composition of cells of various types and in the extracellular space (Figure 11B1, Appendix A). HuCD IF was characteristic of several types of neurons (Figure 11C, Appendix A). Localization of HuCD in the *nucleus rotundus* neurons was usually observed in the cytoplasm of cells and, in some cases, cells were labeled completely (Figure 11C1). The scanning in three channels revealed heterogeneous expression patterns of HuCD+ neurons and DAPI-stained nuclei with numerous nucleoli, and also a co-localization of GFP+ and HuCD+ signals in neurons (Figure 11D). In the case of co-localization of the GFP+ and HuCD+ signals, separate GFP+ granules inside the cytoplasm of HuCD+ neurons, as well as a homogeneous double IF of HuCD and GFP, were detected (Figure 11D1). In these cells, two or more DAPI-stained nucleoli were identified (Figure 11D1). The one-way ANOVA for the distribution of immunolabeled cells did not reveal significant intergroup differences in the distribution of GFP+ and HuCD+ cells (Figure 11E). The percentage of GFP+ and GFP+/HuCD+ cells is presented in Appendix A.

### 2.5. Extracerebellar Transduction of GCaMP6m-GFP into the Corpus Geniculatum

The *corpus geniculatum* (CG) is one of the large visual nuclei in the thalamus, which are part of the synencephalon (Figure 12A, pictogram). The DAPI staining revealed three types of nuclei in the CG of juvenile chum salmon, either having a diffuse localization or forming small clusters (Figure 12A). An examination of the CG under a higher magnification showed that most nuclei had numerous nucleoli (Figure 12A1). Multiple GFP expressions in several cell types and, to a lesser extent, in granules were detected in the CG (Figure 12B, Appendix A). Moderate or intense GFP expression were most often observed in the cell cytoplasm or in single intensely labeled cytoplasmic granules; the area around the nucleus was not labeled (Figure 12B, Appendix A). Completely GFP-labeled cells were detected less commonly (Figure 12B). Intense HuCD IF was detected in three types of neurons (Figure 12C, Appendix A). Most HuCD+ cells had a cytoplasmic localization of HuCD, but in larger neurons, the proximal portions of the processes were also labeled (Figure 12C1). The scanning through three channels revealed heterogeneous HuCD and GFP IF patterns, as well as a multinuclear structure of DAPI-conjugated immunonegative cells (Figure 12E, Appendix A). Patterns of co-localization of GFP+ and HuCD+ signals were identified in the cytoplasm of two cell types (Figure 12D1, Appendix A). In cells with co-localization, a homogeneous cytoplasmic or discrete granular pattern of GFP/HuCD immunolocalization was observed (Figure 12D1). The one-way ANOVA for the distribution of immunolabeled cells revealed intergroup differences in the proportions of immunolabeled GFP+ and HuCD+ cells (*p* ≤ 0.05) and HuCD+ and GFP+/HuCD cells (*p* ≤ 0.05) of cell groups (Figure 10E). Equivalent percentages of GFP+ and GFP+/HuCD+ cells were found in the CG (Appendix A).

## 3. Discussion

### 3.1. Phylogenetic Divergence and Functional Heterogeneity of Fish Cerebellar Connections

The fish cerebellum is one of the integrative systems of the fish brain, including the vestibulolateral lobe (granular eminences and the caudal lobe of the cerebellum), the *corpus cerebelli*, and the cerebellar valve [40]. The results of histological studies of the fish cerebellum, in particular, the presence of a granular and molecular layers, as well as an intermediate ganglion layer, including projection Purkinje cells and eurydendroid cells, indicate their structural and functional unity in members of various taxonomic groups (from the phylogenetically most ancient to the advanced Teleostei) [41,42,43]. Currently, there are reports about the intracerebellar connections in fish and other vertebrates, indicating complex structural transformations of these projections over the course of evolution [18,19,42,44]. Vestibulo-cerebellar projections are considered to be the most ancient connections, which receive input fibers from the primary octavo-lateral system [40]. Connections from the *corpus cerebelli* in fish are also largely comparable to those in other vertebrates [16,44,45]. Among the projection systems in fish, ascending climbing fibers, extending from the internal olive, and also numerous mossy-like fibers (input) projected from the spinal cord, in particular, from the sensory nuclei of the *medulla oblongata*, the motor nuclei of the reticular formation, the lateral reticular nucleus [42], and the *locus coeruleus* [18,40]. Visual signaling in the *corpus cerebelli* of fish is known to come from the nuclei of the pretectum, nuclei of the isthmus (*nucleus isthmi*) [18], and the accessory visual system [40]. Nevertheless, since the nuclei of the pretectal complex are the most variable ones in different groups of fish, further studies of the structural organization and hodological properties of the visual nuclei of the thalamus in each specific group are needed. Moreover, the inputs to the *corpus cerebelli* from the nuclei of the lateral valve, as well as some nuclei of the telencephalon, the dorsal tegmental nucleus, and the nuclei of the paracommissural region, were found exclusively in apomorphic fish [41,43]. The cerebellar valve connections are also highly variable; it was found that signals from the internal olive and the *locus coeruleus* come to the valve, while a much weaker input was recorded from the nucleus of the *lateral lemniscus* and the dorsal tegmental nucleus and isthmus [18,19,40]. However, studies showed that all of the above nuclei of the stem are projected directly into the *corpus cerebelli*, thus, confirming the existence of spatial compartmentalization (incoming signals to the body and valve in fish) [18,19,46]. In some studies, it was found that the cerebellar valve connections differ between fish species. Data on the goldfish *Carassius auratus* showed projections into the valve from the primary sensory neurons of the trigeminal nerve to the area of the isthmus, as well as from the granular eminence and pre-eminence nuclei [46].

In mormyrids, the input fibers to the cerebellar valve extend from the preglomerular nuclei and from the *torus semicircularis*. No similar projections were found in other fish species. Thus, they probably represent a species specialization based on the electrosensory system [46,47]. These intraspecies differences associated with the projections of the cerebellar valve indicate the high plasticity of the sensory inputs of the valve compared to the *corpus cerebelli* [40]. In general, the data on the connections of the cerebellum in fish from different phylogenetic groups support the opinion that the cerebellar body is a plesiomorphic part of the cerebellum characteristic of most vertebrates, while the valve is its evolutionarily new synapomorphic part that appears in Actinopterygii [19,48].

The results of the present study are based on the long-term monitoring of a rAAV (GCaMP6m-GFP), in particular, on the investigation of its potential transduction into CNS cells of juvenile chum salmon and the ability to be transported along anterograde and retrograde projections both inside the cerebellum and as part of remote transcerebellar projections. In studies on the fish brain, the distribution of rAAVs is often associated with the emergence of granules with both intracellular and extracellular localization in projection targets [12]. To date, little is known about the infection process of AAVs after attachment to the cell surface. The mechanism of penetration of most non-enveloped viruses is poorly understood. Several different options have been expressed based on morphological, ultrastructural and biochemical studies. However, the ways in which AAV vectors enter cells with subsequent translocation into the nucleus and positioning within the cell, followed by the expression of transgenes, are still not clear. Viral receptors are often involved in determining the range of hosts and specific tissue tropism of the virus. The recently obtained biochemical and genetic data suggest that the cell-surface heparan sulfate proteoglycan (HSPG) serves as a primary receptor for AAV attachment [25]. The fibroblast growth factor receptor [34] and avb5 integrin [35] are also involved as co-receptors or mediators of AAV entry into target cells. Thus, the long-term monitoring of the rAAV spread after a single injection into the cerebellar body of juvenile chum salmon both provided an integrated view of the vector’s intracerebellar distribution and indicated the presence of extracerebellar projections in which rAAV transduction was identified.

### 3.2. Intra- and Extracerebellar Transduction of rAAV in Fish

According to the results of our observations after the injection into the juvenile chum salmon cerebellar body, transduced cells and granules are present in the dorsal, lateral, and basal zones of the cerebellum, which indicates a wide distribution of the vector in the *corpus cerebelli*. Obviously, the distribution of GCaMP6m-GFP within 3 months post-injection occurred both as a result of the diffusion of the vector and as part of the efferent cerebellar projections extending from the EDCs and, probably, PCs. The existence of such projections in fish was confirmed in previous studies [16,45,49,50]. The contralateral exit from the *corpus cerebelli* in fish is assumed to be formed mainly by EDCs projecting into the region of the ventral thalamus and pretectum [16,45]. Some data indicate the existence of motor projections into the nucleus of the medial longitudinal fasciculus, the reticular formation, and the nuclei of the oculomotor complex [19,40,49,51].

The results of studies on transgenic zebrafish using the red fluorescent protein (RFP) Fyntag RFP-T, along with a transneuronal anterograde indicator and wheat germ agglutinin (WGA), showed that WGA (Tg(olig2: EGFP)) expression is observed not only in the PCs, but also in the EDCs in both embryonic and adult fish [15]. The EDCs are the primary efferents of the PCs and are considered as functional equivalents of the cerebellar nuclei in mammals [44]. In contrast, EGFP-fluorescent granular neurons, which are PC afferents, did not express WGA in transgenic Tg (gata1: EGFP) embryos, juveniles, and adults [15].

In studies on zebrafish, WGA immunolabeling revealed PC efferents, but did not reveal afferent projections of anterograde transneuronal transport. WGA, unlike anterograde/retrograde stains, detects neuronal efferent bodies, but often does not label axons and dendrites [52]. In transgenic zebrafish expressing only FyntagRFP-T in the PCs, no WGA+ immunohistochemical signal was detected [15]. Data on efferent projections of the zebrafish PC are largely consistent with the results of cerebellum hodology studies, based on tracer injections, in other fish species [53,54], and also the results of WGA immunolabeling in mice [52], which confirms the high degree of conservatism of the cerebellar connections during evolution.

Our observations showed a high degree of co-localization of HuCD+ and GFP+ in the neurons of the lateral region (Figure 5D1) and, to a lesser extent, in the basal part of the cerebellum (Figure 6D1). Thus, the distribution of the rAAV (GCaMP6m-GFP) in the dorsal, lateral, and basal zones of the cerebellum during long-term monitoring indicated the presence of transgenic cells expressing both the neuron-specific HuCD protein and the GFP protein, which confirms the expression of transgenes in cerebellar neurons. However, the signal intensity in the lateral region exceeded that of the basal region (Appendix A).

Our results are consistent with the data of transgenic studies on zebrafish embryos, in which, at 5 days post-fertilization (dpf), mosaic PC-specific expression of the membrane marker tagRFP-T was detected in cells of the lateral, but not the medial, part of the cerebellar body [15]. These cells in zebrafish extended their afferents to the octave nuclei [45], as primary efferent projections without contact with the EDCs. These projections in the fish cerebellum are known to be the only direct extracerebellar projections of the PCs in vertebrates [45,55].

Our data indicate the functional heterogeneity of the selected rAAV serotype, with the genetically encoded calcium indicator (GECI) of the latest generation GCaMP6, in particular, and its different abilities to transport in different parts of the brain. This explanation is based on previously obtained data, indicating that different serotypes of adeno-associated viruses may include different proteins in the capsid, which, in turn, provides different parameters of their transduction in the brain [8,56]. In particular, some AAV serotypes are transported along neuronal projections of the injected area [8]. The results of these studies suggest that an injection of a transportable vector into the multiprojective region of the brain or, in the case of juvenile chum salmon, into the cerebellum, can lead to a wide distribution of the transgene. This, in turn, can subsequently be practically implemented in the targeted delivery of specific macromolecules for correction, including therapeutic ones, in a large mass of nervous tissue with a single injection.

The studies on transgenic zebrafish made it possible to determine the projection features of EDCs and PCs from different areas of the cerebellum. In particular, EDCs from the caudolateral and caudomedial cerebellum were mainly associated with octave nuclei [15]. These projections were mostly ipsilateral, with few of them crossing over on the contralateral side. According to our data, there was no transgene expression in the projection cells (PC and EDC) of the cerebellum. The immunofluorescent identification of HuCD showed the expression of the neuron-specific Ca-binding protein in both the EDC and PC populations in juvenile chum salmon. Studies on zebrafish using transneuronal WGA mapping and EDC axon tracking have shown that PCs of the caudal cerebellum project onto the neurons in the nucleus of nerve VIII [15].

The use of non-invasive methods in freely behaving animals showed a regional functional organization of the ganglionic layer in the zebrafish cerebellum [15]. These studies demonstrated the Ca^2+^ imaging and optogenetic manipulation in the caudal cerebellar ganglion layer that controls saccadic eye movements. The swimming behavior of zebrafish is controlled by the rostro-medial part of the ganglionic layer, which is associated with the musculoskeletal system of the CNS. Thus, not only efferent, but also afferent projections of the vertebrate cerebellar PCs, are organized into morphological and functional complexes for various purposes. The further use of non-invasive methods may contribute to answering the question as to how the regionalized neural networks are organized and optimized during differentiation for an optimal physiological effect. Thus, not only the input, but also the output of the ganglionic layer of the vertebrate cerebellum, is organized into various functional modules. The question of how the work of such a regional neural network is organized and optimized during brain differentiation to achieve an optimal physiological result can be addressed by combining the neuroimaging methodology with non-invasive methods of transgenic imaging.

The *corpus cerebelli* in juvenile chum salmon, as in other fish species [40], is a multiprojective area of the brain, connected by afferent and efferent projections with the superior areas of the brain stem and synencephalon, as well as with multiprojection areas of the *medulla oblongata* and spinal cord. During the possible development of the chum salmon cerebellum, as in the development of the zebrafish cerebellum, the lateral part of the larval cerebellum can give rise to the caudomedial part of the adult cerebellum [15], as was previously shown for the mouse cerebellum [57]. Thus, the topographic organizations of the cerebellum and its efferents are probably very similar between fish (chum salmon and zebrafish), mice, and humans. This idea is further supported by a comparison of the zebrafish larval efferents with the WGA staining in transient transgenic adult fish expressing WGA/tagRFP-T in subdomains of the PC layer [15]. Thus, the data obtained suggest that the lateral part of the cerebellum in juvenile chum salmon and zebrafish may correspond to the human *flocculus*, and the caudal part of the cerebellum of juvenile chum salmon corresponds to the human caudal *vermis*, while the rostro-medial part of the fish cerebellum (in salmon salmon and zebrafish) probably corresponds to the human *paravermis* of the cerebellum. Therefore, the presence of established projection areas in the juvenile chum salmon cerebellum, as well as outside it, and the identification of transgene expression in them confirm the potential ability of rAAVs to distribute both in intracerebellar and afferent and efferent extracerebellar projections of the cerebellum.

### 3.3. Vector Transduction into Tectum Opticum

*Tectum opticum* is another integrative center of the fish brain with a laminar structure, including up to 15 layers of neurons and a neuropil that receives multimodal input from various brain regions [58,59]. The major areas that have strong directional connections with the *tectum opticum* are the retina, and also various visual centers of the dorsal and ventral thalamus [60,61]. In addition to visual impulses, the *tectum opticum* receives information from the lateral line [61,62], *torus semicircularis* [63,64] and telencephalon [64,65]. Studies on zebrafish have shown that the part of the EDCs from the caudolateral region are projected to the anterior region of the brain in the hypothalamus and *torus semicircularis* in the midbrain [15]. On the contrary, EDCs in the rostromedial part of the *corpus cerebelli* showed completely different efferents. These EDCs extended long ascending axons to the nucleus of the medial longitudinal fascicle, the red nucleus, and the thalamus. In addition, EDCs from the rostromedial region extended long descending axons into the reticular formation. These long-axon projections were mostly bilateral, except for the projections to the red nucleus, which were exclusively contralateral [15]. In *gymnotoid* fishes [66,67] and mormyrids [68], information from electrosensory organs enters the tectum opticum from the *torus semicircularis* [63]. The results of hodological studies have shown that the tectal projections switch through the *isthmus nucleus* [69,70]. Thus, efferent signals from the cerebellum can pass through the *isthmus nucleus* to the *tectum opticum*.

Studies of the rAAV distribution after injection of the vector into the juvenile chum salmon cerebellum revealed the presence of transgenes in bipolar neurons in the latero-caudal lobe of the *tectum opticum* (Figure 6D1). The results of our studies showed the presence of co-localization of GFP and HuCD in a limited type of bipolar elongated tectum cells, adjacent to the reactive neurogenic DAPI-labeled nuclei in chum salmon (Figure 6D1). We believe that the co-localization patterns of GFP and HuCD in a certain type of bipolar cells, adjacent to the areas of induced neurogenesis, are highly specific. Some data [49,50] and the results of our own studies suggest that the transgene expression in this cell type may be associated with its diffuse transportation as part of non-directional projections of the cerebellum to the isthmus and further to the latero-caudal part of the *tectum opticum*. This assumption is also confirmed by the lack of granules, indicating a non-specific, diffuse pattern of GCaMP6m-GFP distribution. Taking into account that the cerebellum is characterized by a complex structural organization and complex connections, a direct intraparenchymal injection of a rAAV results in the local distribution of the rAAV and is useful for the treatment of CNS diseases associated with specific areas of the brain [24]. On the other hand, its delivery to the cerebrospinal fluid (CSF) space by intratectal administration may provide a wider distribution over the CNS. Unfortunately, these modes of AAV administration are invasive and represent a significant risk in clinical practice. Alternatively, intravenous administration of some vectors, such as AAV9 and AAVrh.10, allowed these transporters to hybridize into the blood–brain barrier (BBB), which proved to be an important condition for neuronal and glial transformation [24,71,72]. This landmark discovery led to a number of studies demonstrating the therapeutic efficacy of systemic rAAV administration for diseases affecting large areas of the CNS, including spinal muscular atrophy [73], amyotrophic lateral sclerosis [74], Canavan disease [75], monosialotetrahexosylganglioside (GM1) gangliosidosis [76], and mucopolysaccharidosis III type [77]. Thus, the results of long-term monitoring in fish can contribute to understanding the mechanisms of rAAV distribution in multiprojective brain systems, which, in turn, can affect the practical use of transgenes as a platform for gene therapy in certain human CNS diseases [24].

It is known that tectal impulses are spatially ordered in the form of tangential clusters, in accordance with their projection features [58]. These clusters are not identical in different layers of the *tectum opticum* [40,59]. In particular, numerous perivitricular tectal neurons, belonging to type 14 according to the Meek’s classification [60], have long dendritic processes that spread radially along the *tectum opticum* and have a different morphology, as well as pre- and postsynaptic contacts in the optic fiber layer and other layers of the *tectum opticum*. Thus, at present, little is known about these cells, which, therefore, remain the subject of future research in order to identify the various visual and non-visual sources that may give rise to this cell type in fish. Efferent projections of some type 14 cells are associated with other visual nuclei, such as, in particular, pretectum, thalamus, and *isthmus nucleus* [60], but most of them are local interneurons. The efferent axons of half of the 13 types of *tectum opticum* neurons reach different centers of the reticular formation and *torus semicircularis* [40,78]. Thus, in cyprinids, the periventricular neurons of the *tectum opticum* may have predominantly ascending and, located on the periphery, descending projections. However, this issue has not yet been fully addressed. In *gymnotoid* fishes, the periventricular tectal neurons also have descending projections, while the peripheral tectal neurons project into the rostral afferent pathways [79]. Nevertheless, in *Navadon modestus*, peripheral tectal neurons of types 2, 4, and 5 project into the pretectum [60,80]. The neurons related to the motor efferents are spatially ordered within a single *tectum opticum* hemisphere and, for the most part, have non-overlapping projection areas in latero-caudal directions that control contralateral eye or body movements [40].

The results of modern physiological and optogenetic studies have shown that, for analyzing PC activity during oculomotor reflex acts and other related behaviors in zebrafish, it is necessary to use genetically encoded calcium indicators (GECI). Their use in PCs is a challenge because GECIs are well known to provide only low changes in signal intensity during the registration of Ca^2+^ transients in the PC’s cerebellum [81]. To address this issue, a number of studies tested the available GECIs for their emission properties in PCs [15]. GCaMP2, GCaMP3, GECO1.1, and GECO1.2 provided low sensitivity [82,83,84]. When using GCaMP5G, sufficiently strong amplitudes were recorded to detect individual PCs with fluctuations in the Ca^2+^ level [15,85]. To elucidate the localization of changes in the Ca^2+^ level in zebrafish larvae during the performance of the optokinetic reflex, continuous recording of Ca^2+^ oscillations at low magnification over the entire PC layer at 3.64 Hz was carried out [15]. It was shown that a recording using genetically encoded calcium indicators (GECI) does not allow for the distinguishing of a single PC action potential [81], but makes it possible to identify the areas of cumulative accumulation of PC activity, which, in turn, allows for the revealing of the physiological involvement of PC regions in certain behavior [15].

In our studies on juvenile chum salmon, we also used a latest-generation vector, based on the genetically encoded calcium indicator GCaMP6, to determine intra- and intercellular localization by the IF of the GFP protein. In the juvenile chum salmon tectum, specific intracellular patterns of GFP localization were revealed in cells with bipolar morphology, of which many were co-localized with the neuron-specific HuCD protein. Taking into account the fact that the cytoarchetonic and modular organization of the *tectum opticum* is aimed at separating multimodal impulses and their topographic representation, which ensures the functionality of neurons, in particular, their integrative function, which, in turn, determines the possibilities for identifying localization, as well as providing motor control. In addition, due to the complex somatotopic organization of the *tectum opticum*, its stratified structure, and the presence of 15 types of differentiated neurons in the *tectum opticum* of fish [60], we assume that the transgene expression in the identified zone of the chum salmon *tectum opticum* has a specific pattern. 

### 3.4. Expression of rAAV in the Thalamus and Nuclei of the Pretectum and Epiphysis

Among the nuclei of the thalamus, the pretectum nuclei are the most characterized by the greatest variability. Currently, three main patterns of pretectal organization are known for various teleost groups [86,87,88,89,90]; however, the functional context of such variability is still poorly understood. In transgenic zebrafish, outside the cerebellum, the WGA staining identified efferents of PC secondary projections in the thalamus, preoptic region, hypothalamus, operculum, octave nuclei, inferior olive, reticular formation, red nucleus, and nucleus of the medial longitudinal fasciculus [15].

*Nucleus rotundus* is the largest nucleus of the thalamus that has been described from various fish taxa [40]. The *nucleus rotundus* reaches the largest size in perciform fishes [91,92]; in cyprinids, the *nucleus rotundus* is somewhat smaller [93,94], with its major functional properties being apparently transferred to the nuclei of the preglomerular complex [40,93]. In salmonids, such as *Oncorhynchus masu*, the *nucleus rotundus* occupies the central region of the anterior thalamus [95] and is also referred to as the anterior thalamic nucleus.

Within the juvenile chum salmon *nucleus rotundus*, GFP expression was found in four types of neurons. Other GFP-expressing regions were found in juvenile chum salmon in the dorsal thalamus and epithalamus. In the dorsal thalamus, GFP-expressing cells were present both in neurons and in granules; however, comparative studies showed a limited number of cells with the GFP and HuCD co-localization (Figure 5F). In the epithalamus, a limited number of GFP-expressing cells and numerous granules were also found (Figure 6B1), indicating the distribution of transgenes both by intracellular transport and by diffusion. However, the co-localization of GFP and HuCD co-expressing cells in the epithalamic region was as low as in the dorsal thalamus (Figure 8D1). Within the *corpus geniculatum*, which belongs to the nuclei of the pretectal complex, an intensive expression of GFP in cells and granules was revealed. In the CG, the density of transgene-expressing cells and granules was somewhat higher compared to the following other thalamic cell groups: dorsal thalamic nucleus, epithalamus, and *nucleus rotundus*. A study of co-localization of GFP and HuCD in the CG showed a relatively small proportion of cells with the GFP and HuCD co-localization in neurons.

Thus, it is obvious that the transgene’s distribution in the area of the dorsal thalamus, epithalamus, round nucleus of the *nucleus rotundus*, and the pretectal complex showed a relatively low number of GFP-expressing neurons, which indicates the targeted distribution of the transgene in the thalamo-cerebellar projections. We assume that the distribution of the rAAV upon initial injection from the cerebellar body was targeted at the nuclei of the dorsal thalamic, pretectal, and glomerular regions as part of the plesiomorphic thalamo-cerebellar projections characteristic of most teleosts. Such transportation could be carried out both via direct thalamo-cerebellar projections and indirectly via the *nucleus isthmi*, which explains the presence of transgenic cells in the *tectum opticum*.

The major goal of studying the brain morphology and axonal connections in various groups of vertebrates is to identify the areas to conduct further studies of the brain functions [1]. In particular, the dopaminergic (DA) complex of the mammalian midbrain is very specific in this regard, since its connections, including internal, afferent and efferent, are so numerous and variable that they greatly complicate, rather than clarify, the understanding of the mechanisms by which one can evaluate the contribution of the complex to the overall functioning of the brain [9]. However, the reticular origin of the DA complex of midbrain neurons confirms this structural complexity [9]. The features that distinguish the structures included in the DA-complex of the midbrain from the reticular formation are dense projections of the DA-complex of the midbrain to the deep nuclei of the telencephalon and the cortex. However, a variety of other structures that are associated with, or actually included in, the reticular formation also have large rostral protrusions. The intralaminar nuclei of the thalamus [96] project massively to the striatum and prefrontal cortex, while the lateral hypothalamic-preoptic continuum [97] has significant projections to the deep nuclei of the telencephalon and the cortex. The basal cholinergic nuclei of the forebrain [98] are characterized by powerful projections to the cortex and, to a lesser extent, to the nuclei of the thalamus. The usual depiction of the reticular formation as “diffusely” organized is complemented by the subtlety and versatility with which it controls the interaction of neuroendocrine, autonomic, and somatomotor effectors [99]. However, a close examination of the diffuse structure, in particular, the projections of the intralaminar nuclei of the thalamus to the cortex and striatum, showed that this is not the case [100,101]. On the contrary, the organization was considered to be “diffuse”, including the DA-complex of the midbrain. Despite the elusiveness of its basic organization, the available data undoubtedly allows us to rank this structure as one of the most intricate and elegant ones in the brain [9].

In lower vertebrates and, in particular, fish, such connections of the brain have not yet been sufficiently investigated, which significantly complicates comparative studies of the structure of the brain and features of its evolution in a number of vertebrates. The mesencephalon in fishes is the main integrative center that, along with the cerebellum, performs complex processing and multimodal analysis of sensory and motor signaling [40,102].

However, the features of the afferent–efferent relationships between the fish cerebellum and the mesencephalon and its projections into the reticular formation are poorly understood. In Wulliman’s studies, the connections between various integrative centers of the fish brain were subject to a comparative analysis, and data on the connection between sensory and motor systems in different fish species studied to date were summarized [40].

We found GFP-labeled cells in the posterior tuberculum area, which is the caudal boundary of cell migration during the prosencephalon formation in embryogenesis [40,103,104]. The posterior tuberculum area, according to the available data in the literature, has usually no direct connection with the cerebellum, but the topographic location in the area of the posterior diencephalon brings this area closer to the perivetricular nuclei of the thalamus, which, in turn, have a connection with the posterior tuberculum area [104]. We suggest that the labeling of the posterior tuberculum area did not have reveal direct cerebellar projections, but diffusion of the transgene in the infundibular CSF, which labels the periventricular cell aggregations. We also observed similar patterns of transgene transport in the nuclei of the posterior dorsal thalamus and epithalamus.

We associate the detection of GFP expression in epiphysis cells with the possible transgene’s distribution and with the CSF flow the outer surface of the brain. The direct delivery of the rAAV into the CSF space by intracerebroventricular injection allows its wide distribution in the CNS [24], which is, however, an invasive method of introducing rAAV with significant limitations. In fish, the connections between the cerebellum and the epiphysis have not been described, but fish have epithalamic-epiphyseal connections, as in other vertebrates [40,53]. However, the surface location of GFP-expressing cells in the epiphysis (pineal gland), as well as the low degree of GFP cells and HuCD co-localization, may indicate a diffuse mode of transgene distribution. Thus, the detection of transgene expression in the epithalamic-epiphyseal complex can be associated with both the direct intracellular transport and the consequence of diffuse transgene migration with CSF flow.

## 4. Material and Methods

### 4.1. Experimental Animals

In this study, we used 50 juvenile (1-year-old) individuals of the Pacific chum salmon, *Oncorhynchus keta*, with a body length of 17–20 cm and a weight of 47–56 g. The animals were obtained from the Ryazanovka Experimental Fish Hatchery in 2020. The animals were kept in a tank with aerated fresh water at a temperature of 14 °C and fed once a day. The daily light/dark cycle was 14/10 h. The concentration of dissolved oxygen in the water was 7–10 mg/dm^3^, which corresponded to normal saturation. All experimental manipulations with the animals were in accordance with the rules listed in the Charter of the A.V. Zhirmunsky National Scientific Center of Marine Biology (NSCMB) FEB RAS and the Commission on Biomedical Ethics, Resource Center of the NSCMB FEB RAS, regulating the humane treatment of experimental animals (approval No. 1-240322 from Meeting No. 1 of the Commission on Biomedical Ethics at NSCMB FEB RAS, 24 March 2022).

### 4.2. Injection of Recombinant Adeno-Associated Virus

We used recombinant adeno-associated mouse hippocampal viruses AAV1.Camc2a.GCaMP6f.WPRE.bGHpA (Inscopix, Palo Alto, CA, USA) prepared for imaging. The packaging, purification, and titering of the vectors were performed by Stanford University (Inscopix, Palo Alto, CA, USA). The recombinant vectors were purified by CsCl precipitation; genomic copy titers were determined as described previously by Gao et al. [105]. The exercise titers were concentration-optimized to 1.68E13 µg/mL, which was functionally confirmed by calcium signal imaging in pyramidal neurons in the dorsal hippocampus of the CA1 mouse. 

The animals were anesthetized in a cuvette containing 0.01% ethyl 3-aminobenzoate methanesulfonate (MS222) (Sigma-Aldrich, St. Louis, MO, USA, cat. No. WXBC9102V) for 5 min at room temperature. After anesthesia, 0.2 µL of a solution of recombinant AAV, diluted in 0.1 M PBS (Tocris Bioscience, Minneapolis, MN, USA; cat. no. 5564, batch no: 5) (pH 7.2), was administered to each experimental animal by injection (*n* = 5 for each group). The injections were made into the dorsal part of the cerebellar body using a Hamilton syringe (*n* = 5 for each group), according to the previously described technique [10]. The control group received 0.2 µL of 0.1 M PBS (*n* = 5). Immediately after the injury, the animals were released into the tank for recovery and further monitoring.

### 4.3. Sample Preparation

After the intracranial injection into the dorsal part of the juvenile chum salmon cerebellar body, video monitoring of the changes in the motor and behavioral activity of the fish in the experimental group was carried out for 1 h. After 3 months, the animals were removed from the experiment and euthanized by rapid decapitation. The brain was pre-fixed in a 4% paraformaldehyde solution (PFA, BioChemica, Cambridge, MA, USA; cat. no. A3813.1000; lot 31000997), prepared in 0.1 M PBS. After the pre-fixation, the brain was extracted from the cranial cavity and fixed in the same solution for 4 h at 4 °C. Then, it was washed in a 30% sucrose solution at 4 °C for two days, with seven changes of the solution. Sections were cut on a freezing microtome (Cryo-star HM 560 MB, Thermo Scientific, Waldorf, CA, USA) and placed on poly-L-lysine coated glass slides (Biovitrum, St. Louis Petersburg, Russia).

### 4.4. Immunofluorescent Labeling

To identify the neuron-specific HuCD protein in rAAV-transduced cells in the brain sections, we labeled them with the respective primary mouse antibody from Chemicon (clone: AD2.38; Chemicon Billerica, St. Louis MA, USA), at a dilution of 1:100. The resulting brain sections were pre-incubated in 0.1 M PBS, supplemented with 0.1% Tween-20 (Sigma-Aldrich, St. Louis, MO, USA, cat. no. P9416) for 5 min. Then, they were incubated with 0.1 M PBS, with the addition of 0.3% Triton X-100 (Sigma-Aldrich, St. Louis, MO, USA, Cat no. T8787) for 30 min at room temperature. Afterwards, they were washed 3 times for 5 min with 0.1 M PBS, with the addition of 0.1% Tween-20. Then, the sections were blocked by immersing in 1% bovine serum albumin solution (BSA) (Sigma-Aldrich, St. Louis, MO, USA, Cat no. B6917), supplemented with 0.1% Tween-20 for 2 h at room temperature. The brain sections were incubated with primary antibodies at 4 °C for 48 h, washed three times in 0.1 M PBS + 0.1% Tween-20 for 5 min, incubated with donkey anti-mouse conjugated secondary antibody Alexa 546 (Invitrogen, New York, NY, USA, dilution 1:200), and washed with 0.1 M PBS and 0.1% Tween-20, supplemented with 1% BSA and 0.1% Tween-20 for 3 min. Afterwards, the sections were embedded in a medium containing DAPI Fluoromount-G (Southern Biotech, Birmingham, AL, USA, cat. no. 0100-20). Secondary antibodies Alexa 546 were also used alone, without primary antibodies, as the control.

### 4.5. Microscopy

For visualization and further morphological analysis, the following research-grade motorized inverted microscope, with an attachment for improved contrasting when working with luminescence, was used: Axiovert 200-M equipped with an ApoTome fluorescent module and digital cameras AxioCam MRM and AxioCam HRC (Carl Zeiss, Germany). For a detailed examination of the cerebellum and other parts of the brain in three dimensions in the multispectral mode (with several fluorochromes), and also for the study of co-localization of GFP/HuCD/DAPI, a LSM 780 NLO confocal laser scanning microscope with the high-resolution Airiskan system (Carl Zeiss, Jena, Germany) was used. To study the micrographs of the preparations, the material was analyzed using the Axio Vision software (Carl Zeiss, Germany). Measurements were taken using multiple objectives at 10×, 20×, and 40× magnifications in 10 randomly selected fields of view for each region of interest. The number of immunolabeled cells in the field of view was calculated at a magnification of 400×. The morphometric analysis of the cell body parameters (measurement of the greater and lesser diameters of the neuron soma) was carried out using the Axio Vision microscope software. Counting was performed in 10 randomly selected study areas (1 microscopic field of view was 0.12 mm^2^) at 400× magnification for each fish. The mean value was determined by averaging the obtained proportions from five fish.

### 4.6. Statistical Analysis

Prior to the experiment, we performed an analysis of the distribution based on the variations in the measured parameters in our previous study [10] and found that we needed a group of at least 4 animals to achieve a statistical significance of 95%. To ensure that we achieved a group size of 4 fish and, at the same time, continued using the minimum number of animals required, we selected a total of 5 animals per experimental group. Cell quantification was performed in a separately selected field of view at a magnification of 10× objective and 20× eye-piece. All data analyses were performed using a blind test to reduce experimenter bias. In order to rank the labeled elements by size groups, all the values of the measured cells/nuclei were divided into non-overlapping size groups (Appendix A) and presented as the mean ± standard deviation of the mean (M ± SD). An analysis of variance (ANOVA test) was used to assess the between-group differences, when comparing the number of GCaMP6m-GFP-labeled HuCD-immunopositive cells and the co-localization of these markers. The values at *p* < 0.05 and *p* < 0.01 were considered statistically significant. All quantitative results in the present study were analyzed using the SPSS software (version 16.0; SPSS Inc., Chicago, IL, USA). Quantification of the morphometric data of the IHC-labeled cells was performed using Statistica 12, Microsoft Excel 2010 and STATA (StataCorp. 2012, Stata Statistical Software: release 12).

## 5. Conclusions

The *corpus cerebelli* in juvenile chum salmon is a multiprojective region of the brain, connected via afferent and efferent projections with the higher located regions of the brainstem and synencephalon, as well as the multiprojection regions of the *medulla oblongata* and spinal cord. During the possible process of postembryonic development of the chum salmon cerebellum, the lateral part of the cerebellum of juveniles can give rise to the caudomedial part of the adult cerebellum, which is consistent with the data on the cerebellum in zebrafish and mice. Thus, the topographic organizations of the cerebellum and its efferents are similar between fish (chum salmon and zebrafish), mice, and humans. The distributions of a rAAV, after an injection of the base vector into the cerebellum, have revealed highly specific patterns of transgene expression in bipolar neurons in the latero-caudal lobe of the juvenile chum salmon *tectum opticum*. The distribution of the rAAV in the regions of the dorsal thalamus, epithalamus, *nucleus rotundus*, and pretectal complex indicates the targeted distribution of the transgene in the thalamo-cerebellar projections of chum salmon. The detection of GFP expression in the cells of the epiphysis and posterior tubercle of juvenile chum salmon is probably associated with the spread of the transgene and with the CSF flow the outer surface and ventricles of the brain. The direct delivery of the rAAV into the CSF space by intracerebroventricular administration allows its wide distribution in the CNS. Thus, the presence of the identified projection areas in the juvenile chum salmon cerebellum, as well as outside it, and the identification of transgene expression in them indicate the potential ability of rAAVs to distribute both in intracerebellar and in afferent and efferent extracerebellar projections of the cerebellum.

The obtained data make it possible to use rAAVs as an effective tool for studying intracerebral connections in fish and other vertebrates, which can be used in various evolutionary-morphological, hodological, optogenetic and comparative studies. The detection of the neuronal phenotype of HuCD-expressing transgenic cells indicates the features and directions of intracellular and transneuronal transport of transgenes in the fish brain. As a result of the studies, it was confirmed that genetic targeting in the CNS can be provided by gene dispersion due to long-axon antero- and retrograde projections from a limited area of the brain. However, to date, the issues related to the potential delivery of genetic vectors remote from the injection site require further study of the mechanisms involved in transgenic vector transport to remote brain regions.

## Figures and Tables

**Figure 1 ijms-23-04947-f001:**
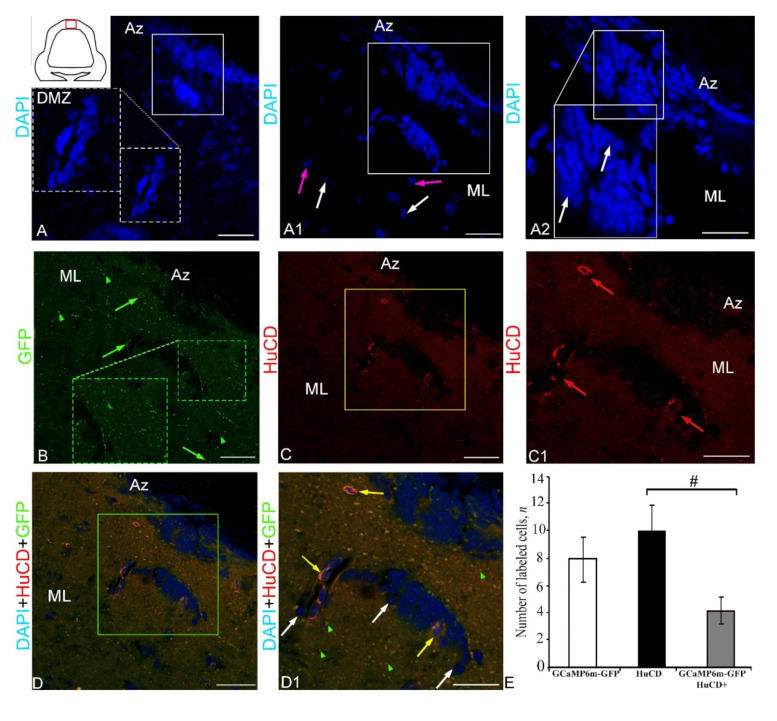
Z-stacks representing DAPI staining, localization of GFP and HuCD, and their co-localization in the cerebellum of juvenile chum salmon, *Oncorhynchus keta*, at 90 days after a single injection of the rAAV into the cerebellar body. (**A**) DAPI staining with a pictogram showing the dorso-medial region of the cerebellar body in a red box, dorsal matrix zone (DMZ) in inset, and apical zone (AZ). (**A1**) A fragment of A3 (outlined by a white square), single rounded DAPI+ nuclei (white arrows), elongated nuclei (pink arrows), and molecular layer (ML). (**A2**) The enlarged fragment of AZ outlined by a white box in A1 and accumulation of heteromorphic nuclei (inset) with numerous nucleoli (white arrows). (**B**) Immunolocalization of green fluorescent protein GFP in granules (green arrowheads) and cells (green arrows) in AZ, accumulation of GFP+ granules (inset) in ML. (**C**) HuCD immunolocalization in AZ neurons (yellow box). (**C1**) The enlarged fragment outlined by a yellow box in C and bodies of HuCD+ neurons (red arrows). (**D**) Co-localization of GFP and HuCD in AZ neurons (green box). (**D1**) The enlarged fragment outlined by a green box in D, DAPI-stained nuclei (white arrows), cytoplasmic colocalization of GFP/HuCD+ neurons (yellow arrows), and GFP+ granules (green arrowheads). Laser scanning confocal microscopy. Scale bar: (**A**) 100 µm, (**A1**,**B**,**C**,**D**) 50 µm, and (**A2**,**C1**,**D1**) 50 µm. (**E**) One-way ANOVA showing the relative distribution of GFP, HuCD, and GFP/HuCD immunolabeled cells of the dorso-medial region of the cerebellum of juvenile *O. keta*; data are presented as mean ± standard deviation (M ± SD). Significant intergroup differences were found in the group of GCaMP6m-GFP and HuCD+ cells (# <0.05) (*n* = 5 in each group).

**Figure 2 ijms-23-04947-f002:**
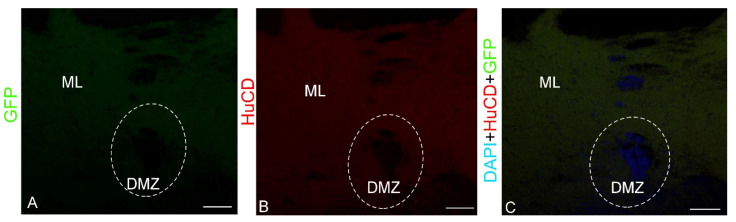
Results of the control immunolabeling experiments for HuCD and GFP in the dorsal part of the cerebellar body in the absence of primary antibodies. (**A**) GFP; (**B**) HuCD; (**C**) DNPI + HuCD + GFP. Dorsal matrix zone (DMZ) is in the white dotted rectangle. ML, molecular layer. Laser scanning confocal microscopy. Scale bar: 100 µm.

**Figure 3 ijms-23-04947-f003:**
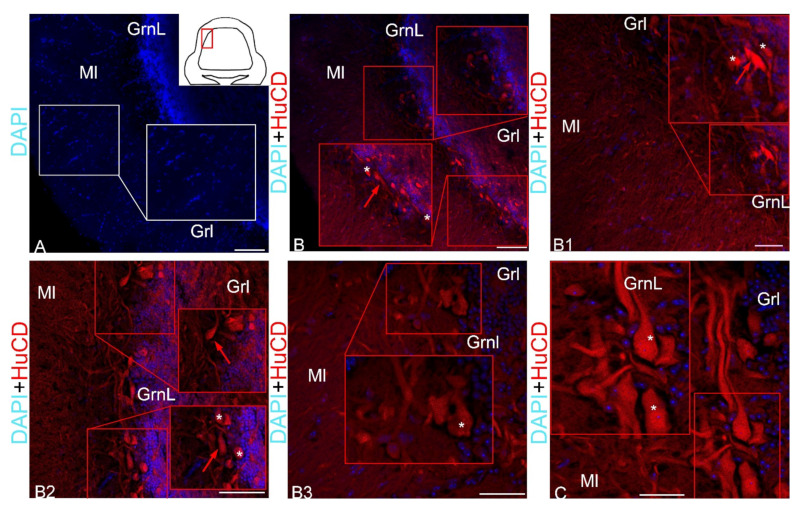
Micrographs showing HuCD immunofluorescence in the ganglionic layer of the lateral region of the cerebellum in juvenile chum salmon, *O. keta*, at 90 days after a single injection of the rAAV into the cerebellar body. (**A**) DAPI staining with a pictogram showing the lateral region of the cerebellum (in a red box), clusters of DAPI-positive nuclei (outlined by a white box); ML, molecular layer; Grl, granular layer; Grnl, ganglionic layer. (**B**) Piriform Purkinje cells (PC) (indicated by a white asterisk) and eurydendroid cells (EDC) (indicated by red arrows in a red box). (**B1**) Intense immunofluorescence of HuCD in PC and EDC; for designations, see (**B**). (**B2**) Intensely labeled PCs and EDCs forming clusters of 2–3 (red inset); for designations, see (**B**). (**B3**) Isolated clusters of HuCD+ PCs and EDCs containing DAPI-stained nuclei and also proximal segments of ascending HuCD+ PC dendrites; for designations, see (**B**). (**C**) Single HuCD+ bipolar EDCs and pear-shaped PCs with proximal HuCD+ dendritic patches; for designations, see (**B**). Laser scanning confocal microscopy. Scale bar: (**A**,**B**) 100 µm; (**B1**,**B2**) 50 µm; (**B3**,**C**) 20 µm.

**Figure 4 ijms-23-04947-f004:**
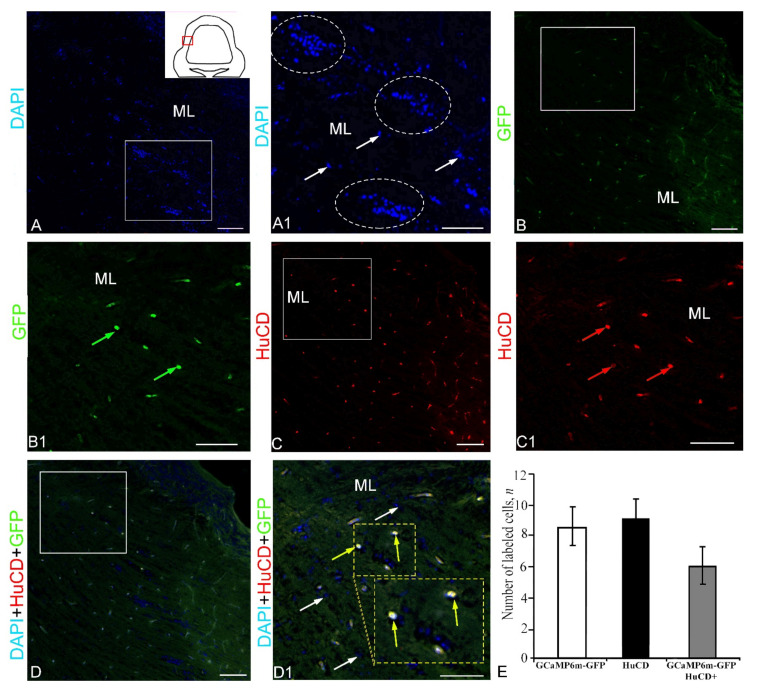
Z-stacks showing DAPI staining, localization of green fluorescent protein GFP and HuCD, and their co-localization in the lateral region of the cerebellum in juvenile chum salmon, *O. keta*, 90 days after a single injection of the rAAV into the cerebellar body. (**A**) DAPI staining with a pictogram showing the lateral region of the cerebellum (in a red box), clusters of DAPI-positive nuclei (outlined by the white box); ML, molecular layer. (**A1**) The enlarged fragment outlined by a white box in A; white dotted line outlines reactive neurogenic zones; single DAPI+ nuclei (white arrows). (**B**) GFP immunolocalization in cells of the lateral zone (in white box). (**B1**) The enlarged fragment outlined by a white box in B (GFP+ cells are indicated by green arrows). (**C**) HuCD immunolocalization in the lateral cerebellum (white box). (**C1**) An enlarged fragment outlined by a white box in C; HuCD+ cells (red arrows). (**D**) Co-localization of GFP and HuCD in cells of the lateral zone (in white box). (**D1**) The enlarged fragment in a white box. (**D**) Cells with co-localization (yellow arrows) are bordered by yellow dotted lines (inset); DAPI+ nuclei (white arrows). Laser scanning confocal microscopy. Scale bar: (**A**,**B**,**C**,**D**) 100 µm; (**A1**,**B1**,**C1**,**D1**) 50 µm; (**E**) One-way ANOVA showing the relative distribution of GFP and HuCD immunolabeled cells (M ± SD) in the lateral region of the cerebellum of juvenile chum salmon, *O. keta*. No significant intergroup differences in GCaMP6m-GFP and HuCD, GCaMP6m-GFP and GCaMP6m-GFP/HuCD+ were found (*n* = 5 in each group).

**Figure 5 ijms-23-04947-f005:**
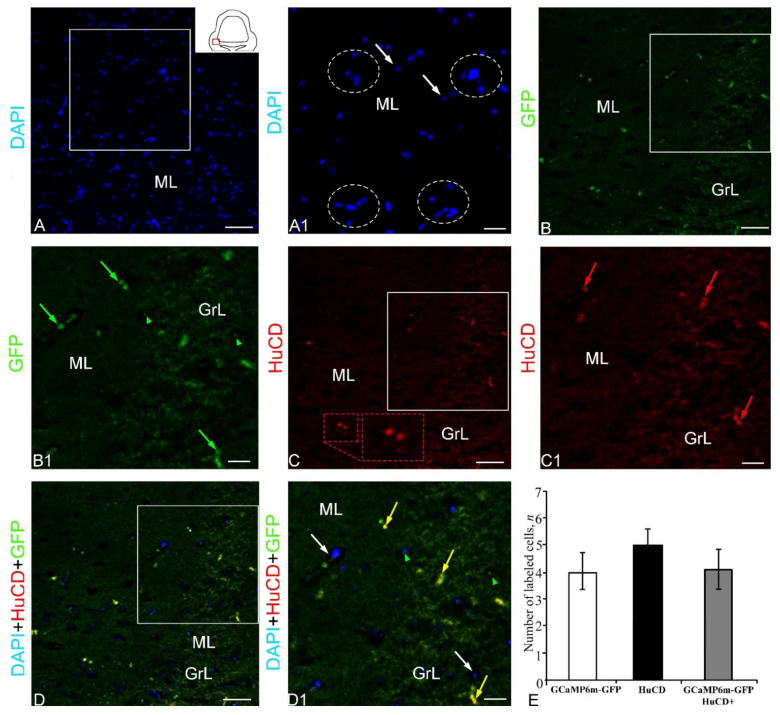
Z-stacks showing DAPI staining, localization of green fluorescent protein GFP and HuCD, and their co-localization in the basal region of the cerebellar body in juvenile chum salmon, *O. keta*, at 90 days after a single injection of the rAAV into the cerebellar body. (**A**) DAPI staining with a pictogram showing the basal region of the cerebellum (in a red box) and a cluster of DAPI-stained nuclei (in a white box); ML, molecular layer. (**A1**) The enlarged fragment outlined by a white rectangle in A; discrete clusters of nuclei and single nuclei (white arrows) are outlined by a white dotted line. (**B**) Expression of the green fluorescent protein GFP in the basal cerebellum (white box); Grl, granular layer. (**B1**) Enlarged view outlined by a white box in B; GFP+ cells (green arrows) and GFP+ granules (green arrowheads). (**C**) Immunofluorescence of HuCD+ neurons in the basal cerebellum (white box); inset (red dotted line) showing enlarged HuCD+ neurons (red dotted inset). (**C1**) The enlarged fragment outlined by a white box in C; HuCD+ neurons (red arrows). (**D**) Optical overlay of three channels DAPI, GFP, HuCD showing areas of GFP/HuCD co-localization in neurons (white box). (**D1**) The enlarged fragment outlined by a white box in D; GFP and HuCD co-localization neurons (yellow arrowheads), DAPI-stained nuclei (white arrowheads), and GFP+ granules (green arrowheads). Laser scanning confocal microscopy. Scale bar: (**A**,**B**,**C**,**D**) 100 µm; (**A1**,**B1**,**C1**,**D1**) 50 µm. (**E**) One-way ANOVA showing the relative distribution of GFP and HuCD immunolabeled cells (M ± SD) in the basal region of the cerebellar body of juvenile chum salmon, *O. keta*. No significant intergroup differences in GCaMP6m-GFP and HuCD, GCaMP6m-GFP and GCaMP6m-GFP/HuCD+ were found (*n* = 5 in each group).

**Figure 6 ijms-23-04947-f006:**
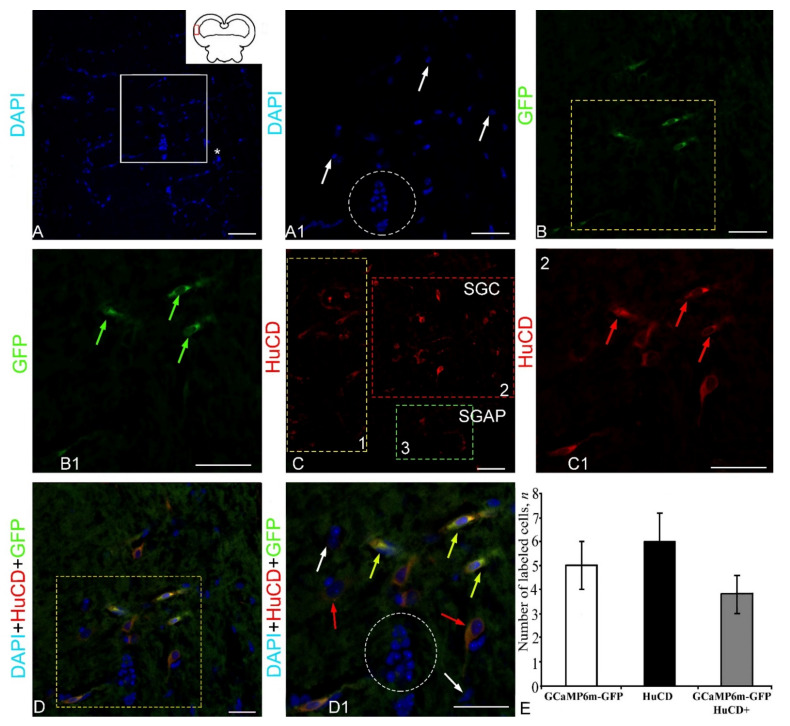
Z-stacks showing DAPI staining, localization of green fluorescent protein GFP and HuCD, and their colocalization in the latero-caudal region of the tectum in juvenile chum salmon, *O. keta*, at 90 days after a single injection of the rAAV into the cerebellar body. (**A**) DAPI staining with a pictogram showing the latero-caudal region of the tectum in a red box; the white box outlines an accumulation of DAPI-stained nuclei in the tectum; ML, molecular layer. (**A1**) The enlarged fragment in A; white dotted line outlines a cluster of nuclei; DAPI+ single nuclei (white arrows). (**B**) Expression of the green fluorescent protein GFP in bipolar neurons (dashed yellow box). (**B1**) The enlarged fragment outlined by a yellow dotted box in B; GFP+ neurons (green arrows). (**C**) Immunofluorescence of the HuCD protein in neuronal populations of *stratum grizerum centrale* (SGC) and *stratum grizerum et album periventriculare* (SGAP). (**C1**) The enlarged fragment outlined by a red dotted box in C; HuCD+ neurons (red arrows). (**D**) Optical overlay of three DAPI, GFP; HuCD channels showing areas of GFP/HuCD co-localization in neurons (dashed yellow box). (**D1**) The enlarged fragment outlined by yellow dotted box in D; neurons with GFP and HuCD co-localization (yellow arrows), DAPI-stained nuclei (white arrows), HuCD+ neurons (red arrows), and a cluster of nuclei (outlined by white dotted oval). Laser scanning confocal microscopy. Scale bar: (**A**,**B**,**C**,**D**) 50 µm; (**A1**,**B1**,**C1**,**D1**) 20 µm. (**E**) One-way ANOVA showing the relative distribution of GFP and HuCD immunolabeled cells (M ± SD) in the latero-caudal region of the tectum in juvenile chum salmon, *O. keta*. No significant intergroup differences in GCaMP6m-GFP and HuCD, GCaMP6m-GFP and GCaMP6m-GFP/HuCD+ were found (*n* = 5 in each group).

**Figure 7 ijms-23-04947-f007:**
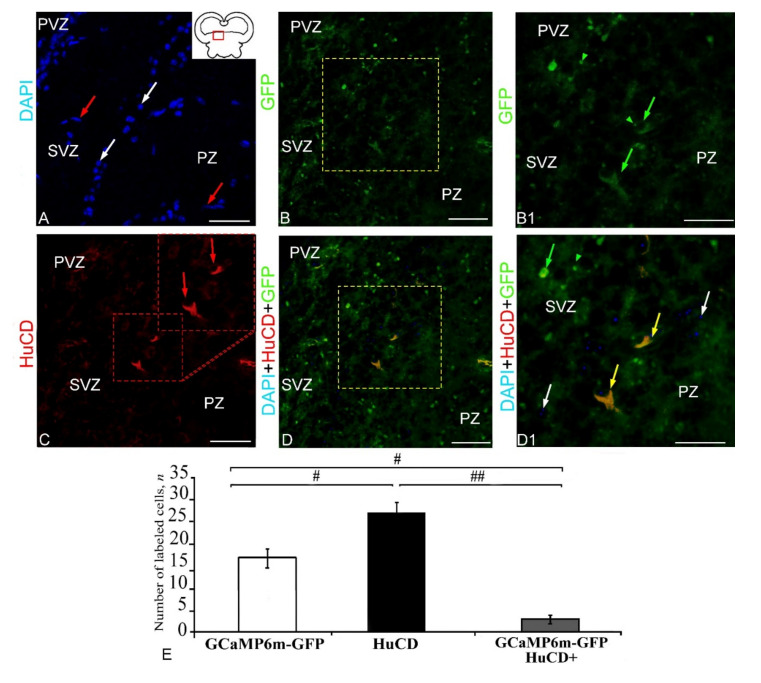
Z-stacks showing DAPI staining, localization of green fluorescent protein GFP and HuCD, and their colocalization in the dorsal thalamus of juvenile chum salmon, *O. keta*, at 90 days after a single injection of the rAAV into the cerebellar body. (**A**) DAPI staining with a pictogram showing the dorsal thalamic region (in a red box), rounded nuclei (white arrows), elongated nuclei (red arrows); PVZ, periventricular zone; SVZ, subventricular zone; PZ, parenchymal zone. (**B**) Expression of green fluorescent protein GFP in dorsal thalamic cells (yellow dotted square). (**B1**) The enlarged fragment outlined by a dotted square in B1; GFP+ neurons (green arrowheads); GFP+ granules (green arrowheads). (**C**) HuCD immunofluorescence in dorsal neurons (red arrows) of thalamus (red dotted inset). (**D**) Optical overlay of three DAPI/GFP/HuCD staining channels showing areas of GFP/HuCD colocalization in neurons (yellow dotted box). (**D1**) The enlarged fragment outlined by a yellow dotted box in D; neurons co-expressing GFP and HuCD (yellow arrows), DAPI-stained nuclei (white arrows), GFP+ neurons (green arrows), and GFP+ granules (green arrowheads). Scanning confocal microscopy. Scale bar: (**A**,**B**,**C**,**D**) 50 µm; (**B1**,**D1**) 20 µm. (**E**) One-way ANOVA showing the relative distribution of GFP and HuCD immunolabeled cells (M ± SD) in the dorsal thalamus of juvenile chum salmon, *O. keta*. Significant intergroup differences were found in GCaMP6m-GFP and HuCD, GCaMP6m-GFP and GCaMP6m-GFP/HuCD+ (# <0.05) (*n* = 5 in each group), and HuCD and GCaMP6m-GFP/HuCD cells (## <0.01) (*n* = 5 in each group).

**Figure 8 ijms-23-04947-f008:**
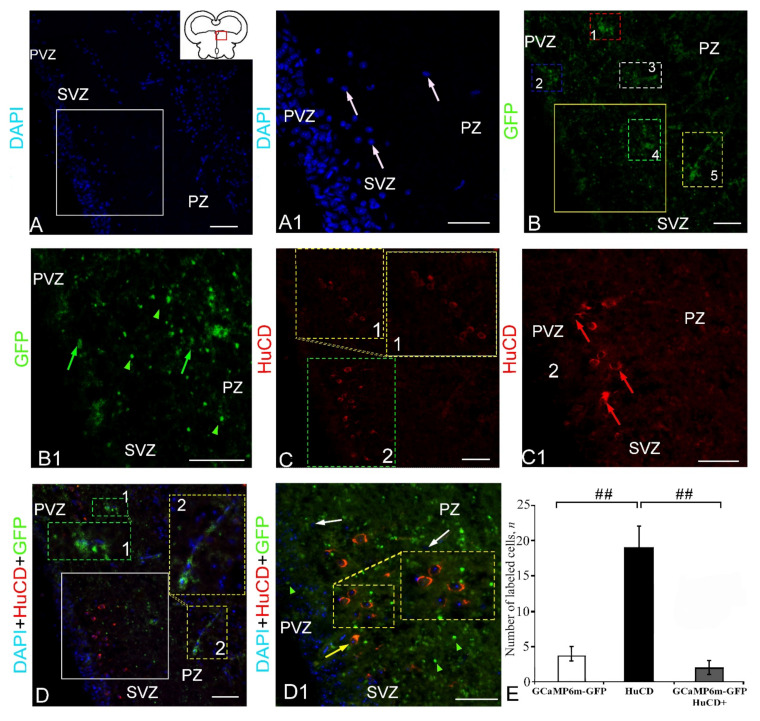
Z-stacks showing DAPI staining, localization of green fluorescent protein GFP and HuCD, and their co-localization in the epithalamus of juvenile chum salmon, *O. keta*, at 90 days after a single injection of the rAAV into the cerebellar body. (**A**) DAPI staining with a pictogram showing an area of epithalamus (in a red box); white box outlines the subventricular aggregation of DAPI-stained nuclei; PVZ, periventricular zone; SVZ, subventricular zone; PV, periventricular zone. (**A1**) The enlarged fragment outlined by a white box in A; heteromorphic DAPI-stained nuclei with nucleoli (white arrows). (**B**) GFP expression in populations of epithalamic neurons (1–5). (**B1**) The enlarged fragment outlined by a yellow box in B; GFP+ neurons (green arrows) and GFP+ granules (green arrowheads). (**C**) Immunofluorescence of the HuCD protein in populations of epithalamic neurons; the yellow dotted inset is an enlarged fragment of population 1. (**C1**) An enlarged fragment of population 2 (green box in C); HuCD+ neurons (red arrows). (**D**) Optical overlay of three DAPI/GFP/HuCD staining channels showing areas of GFP/HuCD co-localization in neurons (yellow dotted box). (**D1**) The enlarged fragment outlined by a yellow dotted box in D; neurons co-expressing GFP and HuCD (yellow arrows), DAPI-stained nuclei (white arrows), GFP+ neurons (green arrows), GFP+ granules (green arrowheads), population 1 co-localization areas (green inset), and population 2 (yellow inset). (**D1**) The enlarged fragment outlined by a white box in D, showing DAPI/GFP/HuCD co-localization in neurons (dashed yellow inset); single cell with colocalization (yellow arrow), DAPI-stained nuclei (white arrows), and GFP+ granules (green arrowheads). Laser scanning confocal microscopy. Scale bar: (**A**,**B**,**C**,**D**) 100 µm; (**A1**,**B1**,**C1**,**D1**) 50 µm. (**E**) Results of the one-way ANOVA showing the comparative distribution of GFP and HuCD immunolabeled cells (M ± SD) in the epithalamus of juvenile chum salmon, *O. keta*. Significant intergroup differences were found in the GCaMP6m-GFP and HuCD groups, and also in HuCD and GCaMP6m-GFP/HuCD (## <0.01) (*n* = 5 in each group).

**Figure 9 ijms-23-04947-f009:**
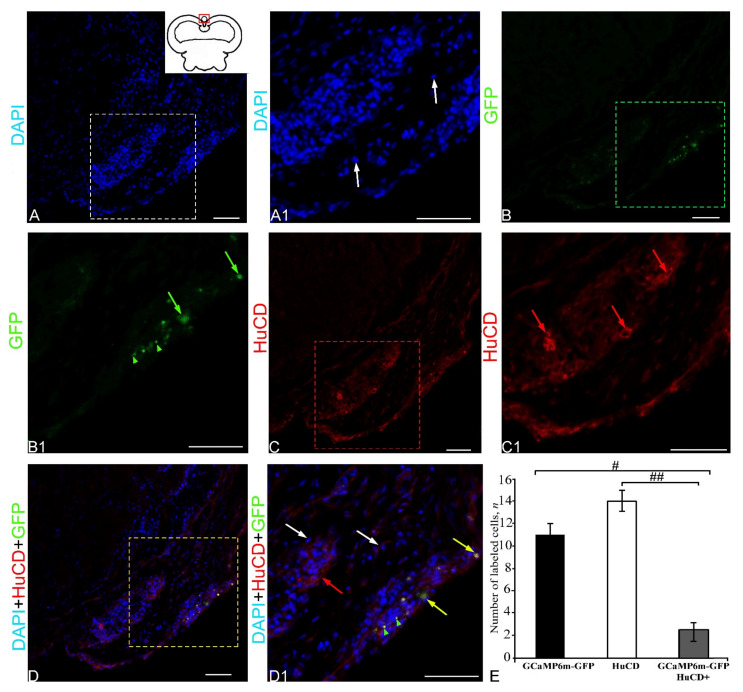
Z-stacks showing DAPI staining, localization of green fluorescent protein GFP and HuCD, and their colocalization in the epiphysis of juvenile chum salmon, *O. keta*, at 90 days after a single injection of the rAAV into the cerebellar body. (**A**) DAPI staining with a pictogram showing the area of the epiphysis (in a red box); the dotted white box outlines a cluster of DAPI-stained nuclei. (**A1**) The enlarged fragment outlined by a white dotted box in A; single DAPI-stained nuclei (white arrows). (**B**) Expression of the green fluorescent protein GFP in pineal cells (green dotted square). (**B1**) The enlarged fragment outlined by green dashed square in B; arrows indicate GFP+ neurons; GFP+ granules (green arrowheads). (**C**) Immunofluorescence of HuCD in pineal cells (in a red dashed square). (**C1**) The enlarged fragment (in a red dashed square) in C; HuCD+ cells (red arrows). (**D**) Optical overlay of three DAPI/GFP/HuCD staining channels showing areas of GFP/HuCD co-localization in pineal cells (yellow dotted box). (**D1**) The enlarged fragment outlined by a yellow dotted box in D; cells co-expressing GFP and HuCD (yellow arrowheads), DAPI-stained nuclei (white arrowheads), HuCD+ cells (red arrowheads), and GFP+/HuCD+ granules (green arrowheads). Laser scanning confocal microscopy. Scale bar: (**A**,**B**,**C**,**D**) 200 µm; (**A1**,**B1**,**C1**,**D1**) 50 µm. (**E**) One-way ANOVA showing the relative distribution of GFP and HuCD immunolabeled cells (M ± SD) in the epiphysis of juvenile chum salmon, *O. keta*. Significant intergroup differences were found in the GCaMP6m-GFP and GCaMP6m-GFP/HuCD (# <0.05) (*n* = 5 in each group) and HuCD and GCaMP6m-GFP/HuCD (## <0.01) groups (*n* = 5 in each group).

**Figure 10 ijms-23-04947-f010:**
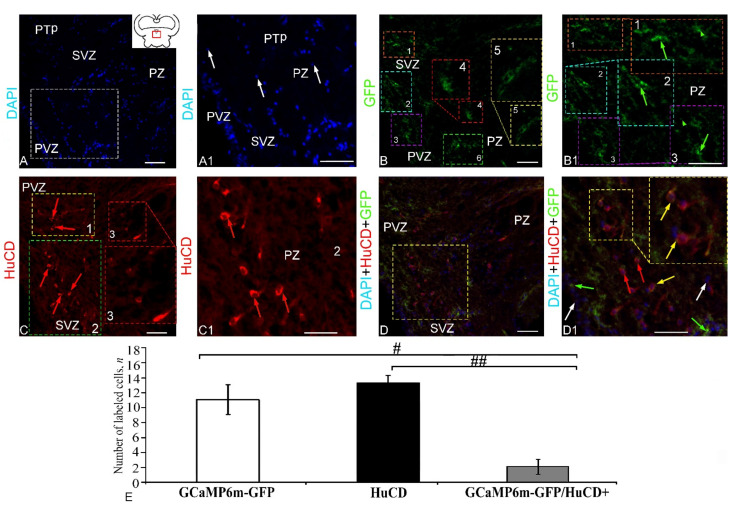
Z-stacks showing DAPI staining, localization of GFP and HuCD, and their co-localization in the posterior tubercle of juvenile chum salmon, *O. keta*, at 90 days after a single injection of the rAAV into the cerebellar body. (**A**) DAPI staining with a pictogram showing the *posterior tuberculum pars posterium* (PTp) area in a red box; a dotted white box outlines the accumulation of DAPI-stained posterior tuberculum nuclei; PVZ, periventricular zone; SVZ, subventricular zone; PV, periventricular zone. (**A1**) The enlarged fragment outlined by a white dotted box in A; stained nuclei (white arrows). (**B**) Expression of the green fluorescent protein GFP in populations of neurons (1–6) of the posterior tuberculum. Insets 4 and 5 show the enlarged fragments of the corresponding populations. (**B1**) The enlarged fragments of populations 1, 2 and 3 (insets) in B; GFP+ cells (green arrows) and GFP+ granules (green arrowheads). (**C**) Immunofluorescence of HuCD in neuronal populations 1 and 2 (red arrows), population 3 (red inset). (**C1**) The enlarged fragment of population 2 inn C; HuCD+ cells (red arrows). (**D**) Optical overlay of three DAPI/GFP/HuCD staining channels showing areas of GFP/HuCD co-localization in neurons (dashed yellow square). (**D1**) The enlarged fragment outlined by a yellow dotted box in D, showing co-localization of GFP and HuCD (yellow arrows in inset), HuCD+ neurons (red arrows), DAPI-stained nuclei (white arrows), and GFP+ neurons (green arrows). Laser scanning confocal microscopy. Scale bar: (**A**,**B**,**C**,**D**) 100 µm; (**A1**,**B1**,**C1**,**D1**) 50 µm. (**E**) One-way ANOVA showing the relative distribution of GFP and HuCD immunolabeled cells (M ± SD) in the posterior tuberculum region of juvenile chum salmon, *O. keta*. Significant intergroup differences were found in GCaMP6m-GFP and GCaMP6m-GFP/HuCD (# <0.05) (*n* = 5 in each group) and in HuCD and GCaMP6m-GFP/HuCD (## <0.01) groups (*n* = 5 in each group).

**Figure 11 ijms-23-04947-f011:**
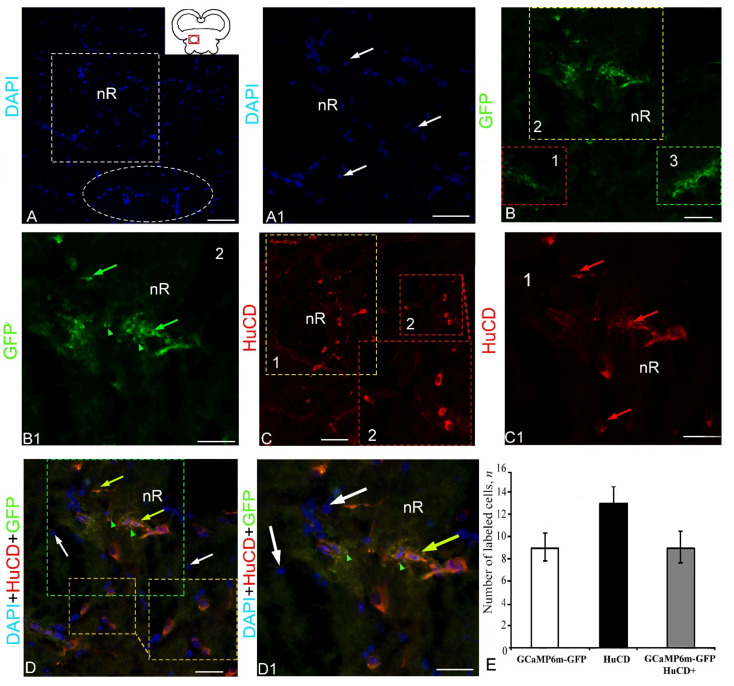
Z-stacks showing DAPI staining, localization of GFP and HuCD, and their colocalization in the *nucleus rotundus* of juvenile chum salmon, *O. keta*, at 90 days after a single injection of the rAAV into the cerebellar body. (**A**) DAPI staining with a pictogram showing the *nucleus rotundus* area in a red box; the dotted white square outlines a cluster of DAPI-stained nuclei; the white dotted oval outlines the fragments of DAPI-stained vessels. (**A1**) The enlarged fragment outlined by a white dotted box in A; DAPI-stained nuclei (white arrows); nR, *nucleus rotundus*. (**B**) Expression of GFP in the populations (1–3) of neurons and *nucleus rotundus* granules. (**B1**) The enlarged fragment outlined by a yellow dotted box in B; GFP+ neurons (green arrows) and GFP+ granules (green arrowheads). (**C**) Immunofluorescence of HuCD protein in populations of *nucleus rotundus* neurons; inset for population 2 with a red dotted line. (**C1**) The enlarged fragment outlined by a yellow dotted oval in C; HuCD+ neurons (red arrows). (**D**) Optical overlay of three DAPI/GFP/HuCD staining channels showing areas of GFP/HuCD co-localization in neurons (yellow arrows) and in GFP+ granules, DAPI-stained nuclei (white arrows); enlarged neurons in the population are outlined by yellow dotted lines (inset). (**D1**) The enlarged fragment outlined by a green dotted oval; for designations, see (**D**). Laser scanning confocal microscopy. Scale bar: (**A**,**B**,**C**,**D**) 50 µm; (**A1**,**B1**,**C1**,**D1**) 20 µm. (**E**) One-way ANOVA showing the relative distribution of GFP and HuCD immunolabeled cells (M ± SD) in the *nucleus rotundus* in juvenile chum salmon, *O. keta*. No significant intergroup differences in GCaMP6m-GFP and HuCD, GCaMP6m-GFP and GCaMP6m-GFP/HuCD+ were found (*n* = 5 in each group).

**Figure 12 ijms-23-04947-f012:**
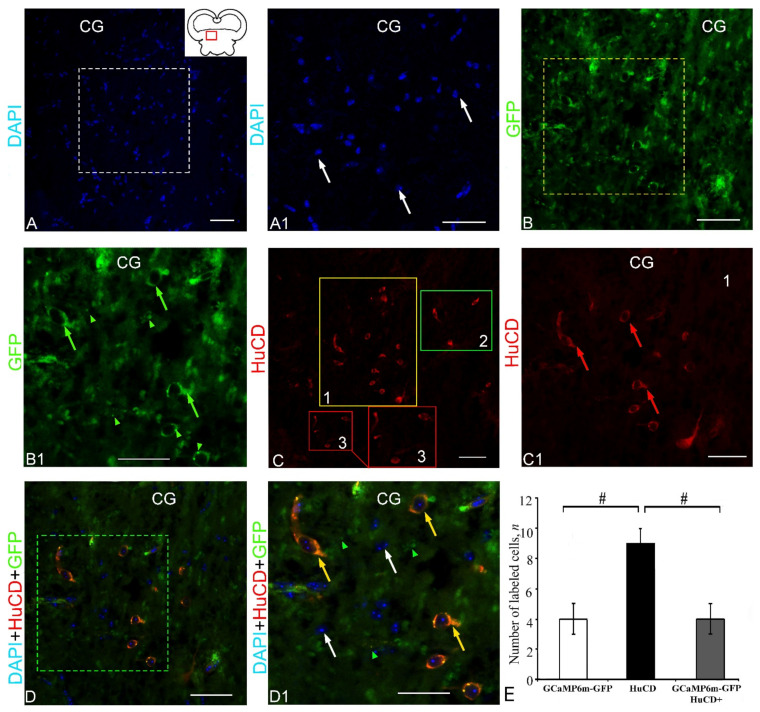
Z-stacks showing DAPI staining, localization of green fluorescent protein GFP and HuCD, and their co-localization in the *corpus geniculatum* of juvenile chum salmon, *O. keta*, at 90 days after a single injection of the rAAV into the cerebellar ody. (**A**) DAPI staining with a pictogram showing the geniculate region (in a red box); the white box outlines a cluster of DAPI-stained nuclei; CG, *corpus geniculatum*. (**A1**) The enlarged fragment outlined by a white dotted box in A; DAPI-stained heteromorphic nuclei with nucleoli (white arrows). (**B**) Expression of green fluorescent protein GFP in geniculate cells (in dotted box). (**B1**) The enlarged fragment outlined by a yellow dotted box in B; GFP+ cells (green arrows) and GFP+ granules (green arrowheads). (**C**) Immunofluorescence of the HuCD protein in populations (1–3) of geniculate neurons. For population 3, the inset shows an enlarged fragment. (**C1**) The enlarged fragment outlined by a yellow box in C; HuCD+ neurons (red arrows). (**D**) Optical overlay of three DAPI/GFP/HuCD staining channels showing areas of GFP/HuCD co-localization in neurons (green dotted box). (**D1**) The enlarged fragment outlined by a green dotted box in D; cells co-expressing GFP and HuCD (yellow arrowheads), DAPI-stained nuclei with nucleoli (white arrowheads), and GFP+ granules (green arrowheads). Scale bar: (**A**,**B**,**C**,**D**) 50 µm; (**A1**,**B1**,**C1**,**D1**) 20 µm. (**E**) Results of one-way ANOVA showing the relative distribution of GFP and HuCD immunolabeled cells (M ± SD) in the *corpus geniculatum* region of juvenile chum salmon, *O. keta*. Significant intergroup differences were found in the GCaMP6m-GFP and HuCD groups and also in HuCD and GCaMP6m-GFP/HuCD (# <0.05) (*n* = 5 in each group).

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
