# Peer review of "Transduction of Brain Neurons in Juvenile Chum Salmon (Oncorhynchus keta) with Recombinant Adeno-Associated Hippocampal Virus Injected into the Cerebellum during Long-Term Monitoring"

_ijms, 2022, doi:10.3390/ijms23094947_

Round 1
Reviewer 1 Report
The manuscript ijms-1692665 reports experimental data on the ability of recombinant adeno-associated viral vectors to distribute themselves in the intracerebellar and extracerebellar afferent and efferent projections of the cerebellum, using Oncorhynchus keta as an experimental model.
The topic is extremely interesting, the results are new and increase knowledge on the development and functionality of the nervous system. The manuscript is well organized and well written. I only found a few typos in the text, easily fixed with a quick review. The discussion is very well argued.
I believe this manuscript is appropriate for publication in the International Journal of Molecular Sciences and I suggest accepting it after a little review.
In particular, statistical analysis must be slightly implemented, with an additional test to the pairwise analysis of variance, such as the Student-Newman-Keuls test (but there are other similar tests that the authors can apply). This is necessary because there are more than two experimental groups to compare.
Furthermore, the Authors should add something in the conclusion section regarding the scientific implications of their findings and possible future perspectives.
Author Response
Dear Reviewer, we thank You for your attention and precious recommendations related to our work.
- Regarding your recommendation to conduct additional statistical analysis, in this work the main goal was to estimate the variance between groups of cells labeled with GCaMP6m-GFP, HuCD and GCaMP6m-GFP/HuCD. This was done for each region of the brain in which transgene expression was detected. However, analysis of variance showed that not all brain structures labeled with GCaMP6m-GFP, HuCD, as well as GCaMP6m-GFP/HuCD cololiza- tion had significant intergroup differences. For example, for the lateral zone (Fig. 4E), the basal zone (Fig. 5E) of the cerebellum, the latero-caudal zone of the tectum (Fig. 6E), and the nucleus rotundus (Fig. 11E), there were significant differences between groups of cells labeled with GCaMP6m-GFP, HuCD and GCaMP6m-GFP/HuCD were not detected. Therefore, the results of the distribution of labeled cells in these figures were previously presented as the Mean ± Standard deviation. Appropriate corrections have been made to these figures. In other areas of the brain: dorsal thalamus, epithalamus, epiphysis, posterior tubercle and corpus geniculatum (Fig. 7,8,9,10, 12), on the contrary, the results of analysis of variance revealed significant intergroup differences in the distribution of labeled GCaMP6m-GFP, HuCD and GCaMP6m -GFP/HuCD cells with different levels of significance (p≤0.05 and p≤0.01). To determine the base factor, which could be perceived as the main group, with which other groups could be compared (using the Student-Newman-Keuls test), it is necessary to conduct a multivariate analysis of variance, which we have not yet performed because we do not have relevant data.
- Information has been added to the conclusion section regarding the scientific implications of their findings and possible future prospects.

Reviewer 2 Report
This is a comprehensive study investigating cerebellum connections in the juvenile salmon with a rAAV approach. The manuscript is well written, the results well illustrated and the discussion comprehensive. Without daubt, the data of this study is of interest for the field.
I only have several minor comments:
Line 15: What means "s"?
Line 25: Not sure whether "washing" is the best term here.
Line 40: ...delievers for example nerve growth factors...
Line 86: Is "optogenetic diagnostics" an established term? If not, consider rephrasing.
Line 93: ...to treat Parkinson's disease...
Figures: In general, I had difficulties to immediately understand how the different microscopic images and their inlays were related to each other. However, I have to admit that I also don't know an easier way and on the end, I also understood the relations well. It is a good idea to indicate with value the relationship between the images but why not using the panel caption if the image has anyway a caption. For example: Fig. 8C has rectangle "2" which could be named "C1". Btw, sometimes (like in this case), the rectangles do not correspond to the magnification, i.e., the green rectangle in Fig. 8C should be a square.
In line 791 & 839, something went wrong with the font size of "gymnotoids". The same happened in the methods section (l. 1019, 1030).
Author Response
Dear Reviewer, we thank you for your attention and important recommendations related to our work.
- The corrections recommended by the Reviewer were made in the corresponding lines of the text, and the editorial revision of the entire text of the manuscript was also carried out.
- We took into account your recommendations regarding changing the rectangular insets to square ones in fig. (Figure 1D; Figure 6A, 6B, 6D; Figure 7B, 7D; Figure 8A, 8B, 8D; Figure 9C; Figure 10A, 10D; Figure 12A, 12B, 12D). However, in Figure 6C; Figure 8C; Figure 10C; Figure 11C; Figure 12C. the original rectangle was retained because we used part of the cell population in a square inset to display the general pattern of cell distribution at low magnification to make the montage. New versions of the drawings are presented in the corrected version of manuscript and attached files.
